JCB Journal of Cell Biology

## TOOLS

# Identification of small molecule inhibitors of G3BP-driven stress granule formation

Brian D. Freibaum[1], James Messing[1], Haruko Nakamura[1], Ugur Yurtsever[1], Jinjun Wu[1], Hong Joo Kim[1], Jeff Hixon[2], Rene Marc Lemieux[2], Jay Duffner[2], Walter Huynh[2], Kathy Wong[2], Michael White[2], Christina Lee[2], Rachel E. Meyers[2], Roy Parker[3,4], and J. Paul Taylor[1,4]

**Stress granule formation is triggered by the release of mRNAs from polysomes and is promoted by the action of the RNA-binding proteins G3BP1/2. Stress granules have been implicated in several disease states, including cancer and neurodegeneration. Consequently, compounds that limit stress granule formation or promote their dissolution have potential as both experimental tools and novel therapeutics. Herein, we describe two small molecules, G3BP inhibitor a and b (G3Ia and G3Ib), designed to bind to a specific pocket in G3BP1/2 that is targeted by viral inhibitors of G3BP1/2 function. In addition to disrupting the co-condensation of RNA, G3BP1, and caprin 1 in vitro, these compounds inhibit stress granule formation in cells treated prior to or concurrent with stress and dissolve pre-existing stress granules. These effects are consistent across multiple cell types and a variety of initiating stressors. Thus, these compounds represent powerful tools to probe the biology of stress granules and hold promise for therapeutic interventions designed to modulate stress granule formation.**

## Introduction

Biomolecular condensation underlies the formation of diverse non-membrane-bound intracellular assemblies, including a wide variety of ribonucleoprotein (RNP) granules (Banani et al., 2017). Biomolecular condensation can occur through liquid–liquid phase separation (LLPS), a process wherein macromolecular components of a mixed phase demix to produce two spatial regions: a higher density, condensed phase containing high concentrations of macromolecules, and a lower density, dilute phase that contains lower concentrations of these macromolecules (Shin and Brangwynne, 2017).

RNP granules are one type of biomolecular condensate that can be found throughout the cell, including the nucleus (e.g., nucleoli, Cajal bodies, speckles), cytoplasm (e.g., P bodies, stress granules), and neuronal processes (e.g., RNA transport granules; Hirose et al., 2022). Composed of RNA and protein, these condensates vary widely in size and cellular function, with roles that include RNA metabolism, ribosome biogenesis, and signal transduction (Banani et al., 2017). On a broader level, the complex and intersecting protein–RNA networks that govern the formation of these condensates are key determinants of intracellular organization (Banani et al., 2017; Sanders et al., 2020; Yang et al., 2020).

Stress granules are a specialized type of cytoplasmic RNP granule that have been widely used as an archetypal RNP granule to uncover fundamental principles of biomolecular condensation (Molliex et al., 2015; Protter and Parker, 2016). Stress granules have been implicated in several disease states, including cancer, where they may promote tumor chemoresistance (Gao et al., 2019; Khong et al., 2022 Preprint; Lavalée et al., 2021), and neurodegeneration, where aberrant granule condensation may lead to pleotropic cellular defects and accrual of proteinaceous deposits (Kim et al., 2013; Mathieu et al., 2020; Molliex et al., 2015; Wolozin and Ivanov, 2019; Zhang et al., 2018). Thus, targeting stress granules holds promise for therapeutic purposes (Patel et al., 2022).

Stress granules undergo an ordered assembly and disassembly process driven by a combination of protein–protein, protein–RNA, and RNA–RNA interactions (Millar et al., 2023; Van Treeck and Parker, 2018) within a defined RNA-protein network (Yang et al., 2020). The most central node within this network is the RNA-binding protein G3BP1 and its paralog G3BP2 (Kedersha et al., 2016; Tourriere et al., 2023; Yang et al., 2020). Depletion of G3BP1/2 from cells eliminates stress granule assembly in response to some stresses (Kedersha et al., 2016; Yang et al., 2020), and disruption of G3BP1/2 can result in decondensation (Gwon et al., 2021). For example, ubiquitination of G3BP1 in the context of heat stress enables the selective extraction of G3BP1 from the stress granule network, causing the system to fall below the percolation threshold and disassemble (Gwon et al., 2021). The reverse is also true: enforced assembly

[1]Department of Cell and Molecular Biology, St. Jude Children's Research Hospital, Memphis, TN, USA;  [2]Faze Medicines, Cambridge, MA, USA;  [3]Department of Biochemistry, University of Colorado, Boulder, CO, USA;  [4]Howard Hughes Medical Institute, Chevy Chase, MD, USA.

Correspondence to J. Paul Taylor: jpaul.taylor@stjude.org.

of G3BP1 multimers triggers the formation of stress granules even in the absence of an initiating stress (Zhang et al., 2019), and supplementation of cellular lysates with purified G3BP1 induces concentration-dependent condensation to form granules with the characteristic proteomic and transcriptomic composition of stress granules (Freibaum et al., 2021).

G3BP1 and G3BP2 each exists as a dimer, with each monomer comprising a folded dimerization domain (NTF2L), a folded RNA recognition motif (RRM), and three intrinsically disordered regions (IDRs). Dimerization via the NTF2L domain is necessary for stress granule assembly, and experimental disruption of G3BP1 dimerization results in rapid disassembly of fully formed stress granules (Yang et al., 2020). The NTF2L domain also harbors the binding site for other stress granule proteins, including the RNA-binding proteins caprin 1 and USP10, which respectively promote or limit stress granule assembly (Kedersha et al., 2016). Indeed, the binding of stress granule proteins at this site may serve to coordinate the interaction network that governs the ordered assembly of stress granules (Yang et al., 2020).

Insights into the specific properties of NTF2L binding have emerged from unrelated studies examining the role of G3BP1 as a regulator of viral replication (Ge et al., 2022; Kang et al., 2021; Panas et al., 2015). For example, Chikungunya viruses inhibit stress granule formation through the interaction of the viral nsP3 peptide with a specific binding pocket within the NTF2L domain of G3BP1/2 (Fros et al., 2012). Deeper examination of the nsP3 peptide has identified the core G3BP1/2-binding motif as consisting of two FGDF motifs, in which both phenylalanine and the glycine residue are required for binding (Panas et al., 2015). This viral strategy may be exploited to modulate human disease, as transfection of nsP3 into mammalian cells can disrupt stress granule formation (Lu et al., 2021).

Here, we designed stable molecules to mimic the interaction of the FGDF motif of nsP3 with G3BP1/2 (Kristensen, 2015). We succeeded in developing two molecules, which we refer to as G3BP inhibitor a (G3Ia) and G3BP inhibitor b (G3Ib), that blocked the association of G3BP with its binding partners and inhibited condensation of G3BP both in vitro and in live cells. Treatment of cells with these molecules resulted in the prevention and/or dissolution of stress granules induced either through exogenous stresses or through the expression of disease-causing mutant proteins. These studies demonstrate that the nsP3 binding pocket of G3BP is an effective target to disrupt stress granules in multiple contexts, including a variety of cell types and initiating stressors. Furthermore, the efficacy of our small molecule compounds suggests the potential for this approach both in the context of therapeutics and as tools to enable mechanistic studies of RNP granule assembly.

## Results

### Identification of peptide mimetics that bind G3BP1

Previous investigations into the role of G3BP1 as a regulator of viral infections have demonstrated that stress granule formation can be blocked by the binding of the viral nsP3 peptide to the NTF2L domain of G3BP1 (Fros et al., 2012; Lu et al., 2021; Panas et al., 2015). Guided by these studies, we sought to design a series of related modified peptides that could function as small molecule inhibitors of stress granule formation. Beginning with the nsP3 25-mer peptide, we first narrowed the peptide down to an 8-mer that retained the ability to bind the NTF2L domain of G3BP1. Using this minimal 8-mer region, we synthesized multiple lead compounds, which we screened for their capacity to tightly bind the nsP3 binding site within the NFT2L domain of G3BP1 (Fig. S1). Two such compounds, referred to herein as G3BP inhibitor a and b (G3Ia and G3Ib), bound to G3BP1 with respective $K_d$ values of 0.54 and 0.15 µM binding as assayed by SPR (Fig. 1, A and B). As negative controls, we also synthesized inactive enantiomers (G3Ia′ and G3Ib′), which bound the same domain with $K_d$ values that were >100-fold higher ($K_d$ values of 75.5 and 44.5 µM, respectively). To confirm binding to the FGDF pocket of G3BP, we performed a peptide displacement assay such that FRET activity was lost when a compound displaced a FAM-labeled FGDF peptide from the NTF2L domain of G3BP1. We found that G3Ia and G3Ib potently displaced the FGDF peptide, whereas their corresponding enantiomers had little effect, even at very high doses (Fig. 1 C), consistent with the strongly reduced affinity of the enantiomers for the NTF2L domain of G3BP1 (Fig. 1 B). To better understand the nature of the tight binding between G3Ia and the NTF2L domain, we obtained an x-ray crystal structure that revealed a number of important hydrogen bonds between the compound and N122, K123, and F124 of the protein. In addition, a key phenyl ring in G3Ia extended the pi-stacking network formed by F15 and F33 of the NTF2L domain (Fig. 1 D). Given the promising properties of G3Ia and G3Ib, including their relatively small polar surface areas suggesting that they might be cell-penetrant (Fig. 1 E), we prioritized G3Ia and G3Ib for further exploration of their capacity to inhibit G3BP-dependent phase separation and/or stress granule formation in cells.

### G3Ia and G3Ib disrupt in vitro condensation of RNA, G3BP1, and caprin 1

Binding of the nsP3 peptide to G3BP1 reduces interaction between G3BP1 and its protein partners caprin 1 and USP10 (Kedersha et al., 2016). To test whether G3I compounds have the same effect as nsP3 peptide, we added these compounds to lysates from U2OS cells expressing G3BP1-GFP and used coimmunoprecipitation to assess the interaction between G3BP1 and endogenous caprin 1 and USP10. As predicted, the addition of G3Ib to these lysates resulted in a dose-dependent reduction of the interaction between G3BP1 and both endogenous caprin 1 and endogenous USP10, whereas the inactive enantiomer G3Ib′ had no such effect (Fig. 2, A–D). This dose-dependent reduction in interaction with caprin 1 and USP10 was also observed in lysates expressing the NTF2L domain of G3BP1 (GFP-NTF2L; Fig. 2, E and F).

Caprin 1 is thought to facilitate stress granule assembly by providing additional valency within the multimeric protein-RNA interaction network, thus promoting LLPS and stress granule formation in cells (Sanders et al., 2020; Yang et al., 2020). Indeed, caprin 1, G3BP1, and RNA undergo co-condensation readily in vitro (Song et al., 2022), and adding purified recombinant caprin 1 to a mixture of G3BP1 and RNA

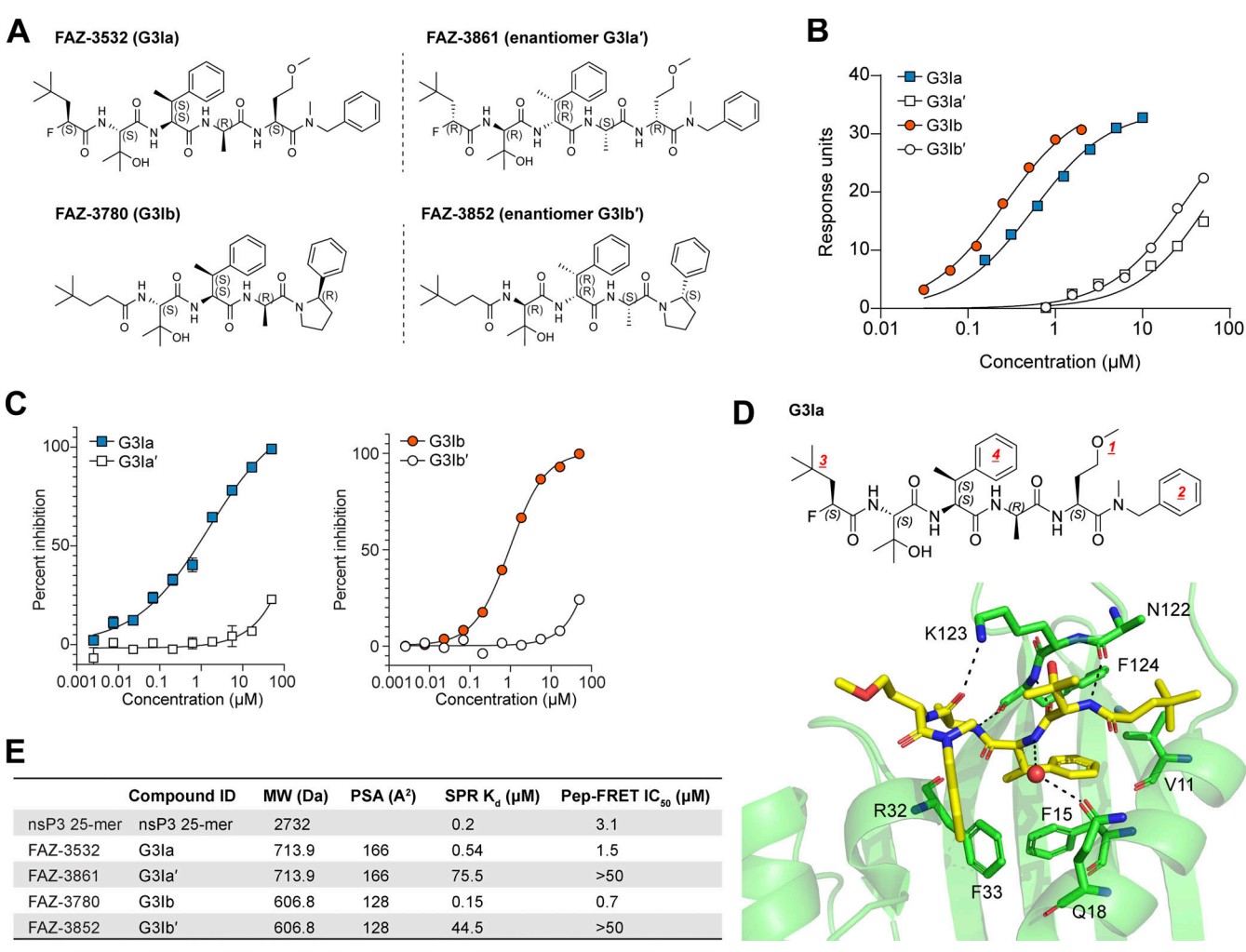

**Figure 1. Lead compounds G3Ia and G3Ib bind with high affinity to the NTF2L nsP3 binding pocket of G3BP1. (A)** Lead compounds FAZ-3532 (G3Ia) and FAZ-3780 (G3Ib) along with respective enantiomer controls FAZ-3861 (G3Ia′) and FAZ-3852 (G3Ib′). **(B)** Representative double-reference subtracted sensorgrams of compounds binding to sensor-immobilized human G3BP1. Compounds were tested by 1/2 dilution with top concentrations of 50 µM (G3Ia′), 50 µM (G3Ib′), 10 µM (G3Ia), and 2 µM (G3Ib). Marks on each curve indicate the time span at which equilibrium binding was measured to estimate the equilibrium dissociation constant using a 1:1 Langmuir binding model. **(C)** Percent inhibition calculated using a peptide displacement assay at indicated doses of G3I compounds. Error bars represent mean ± SD, $n$ = 2 replicates per dose. **(D)** Crystal structure showing the interaction of G3Ia with the nsP3 binding pocket in the NTF2L domain of G3BP1. The NTF2L domain of G3BP1 (light green cartoon model) crystallized in the presence of G3Ia (yellow sticks), with six copies in the asymmetric unit and copy A shown above. All copies were compound bound, although only half had full compound density. The other three were incomplete in either the ether group (1) or terminal phenylalanine (2), highlighting their flexibility. Tert-butyl (3) functions as a space-filling moiety, maximizing the hydrophobicity of the subpocket lined by V11 and F124. An indirect water-mediated backbone interaction with Q18 is present in four of six copies, including copy A above (large red ball). Modified phenylalanine (4) extends the pi-stacking network formed by F15 and F33. **(E)** Summary characteristics of the four G3I compounds compared with the nsP3 25-mer peptide. MW, molecular weight; PSA, polar surface area; SPR, surface plasmon resonance; Pep-FRET, peptide-fluorescence resonance energy transfer.

significantly reduces the threshold concentration of G3BP1 and RNA necessary for LLPS to occur (Yang et al., 2020). To test the effect of our compounds on condensate formation by these molecules in vitro, we added G3I compounds to mixtures of purified recombinant caprin 1, G3BP1, and RNA and found that both G3Ia and G3Ib, but not G3Ia′ or G3Ib′, disrupted the co-condensation of these molecules (Fig. 2 G). These results suggest that active G3I compounds effectively disrupt the binding of G3BP1 with caprin 1 and reduce condensation of G3BP1 with RNA. With this mechanism of action in mind, we hypothesized that condensate formation would be unaffected in a two-component system that consisted of G3BP1 protein and RNA but lacked a binding partner for the NTF2L domain. Indeed, we

found that condensate formation by G3BP1 and RNA was unaffected by the addition of G3Ib, even at very high concentrations (166 µM; Fig. 2 H). In light of these experimental results, the predicted cell permeability of the G3I compounds, and the lack of measurable cell-based toxicity (Fig. 2 I), we pursued these compounds as potential modulators of stress granule formation in living cells.

**Preincubation with G3Ia or G3Ib prevents the formation of stress granules in cells**

To test the effectiveness of G3Ia and G3Ib in blocking stress granule formation in cells, we used U2OS cells stably expressing G3BP1-GFP and examined the formation of stress granules using

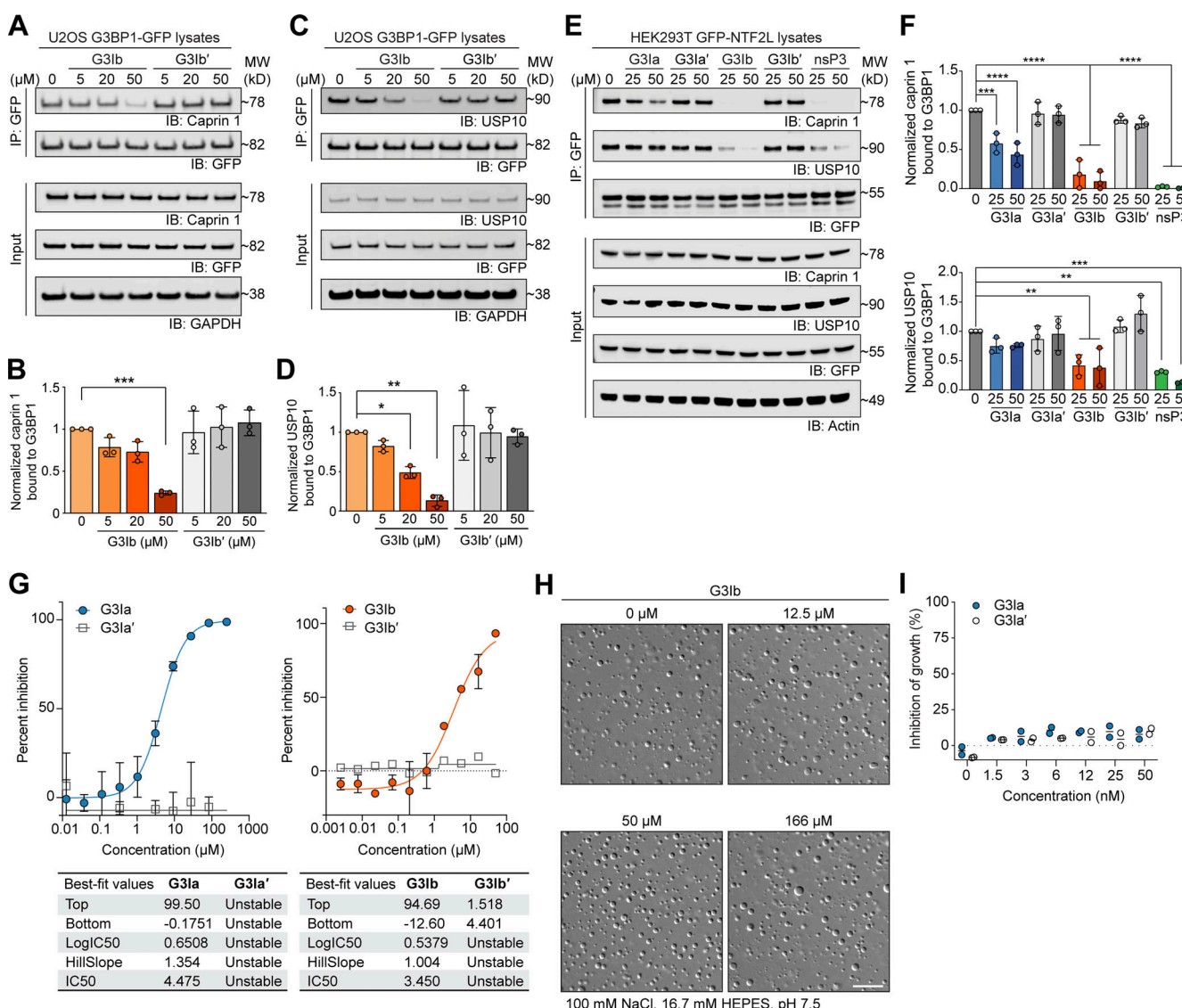

Figure 2. **G3Ia and G3Ib disrupt in vitro condensation of RNA, G3BP1, and caprin 1. (A–D)** Lysates from U2OS cells stably expressing G3BP1-GFP were collected, incubated with increasing concentrations of G3I compound, immunoprecipitated for GFP, and separated by SDS-PAGE. Blots were probed for GFP and endogenous caprin 1 (A and B) or endogenous USP10 (C and D). GAPDH was used as a loading control. Densitometry from n = 3 blots was used to generate graphs (B and D); error bars represent mean ± SD. *P = 0.0495, **P = 0.0011, ***P = 0.0002 by one-way ANOVA with Dunnett's multiple comparisons test. **(E)** Lysates from HEK293T cells expressing GFP-NTF2L were collected, incubated with increasing concentrations of G3I compounds or nsP3, immunoprecipitated for GFP, and separated by SDS-PAGE. Blots were probed for GFP, endogenous caprin 1, and endogenous USP10. Actin was used as a loading control. A representative blot is shown from n = 3 experiments. **(F)** Quantification of densitometry from n = 3 blots as shown in E. Error bars represent mean ± SD. ***P = 0.0006 and ****P < 0.0001 for caprin 1, **P = 0.0087 (25 µM G3Ib), **P = 0.0050 (50 µM G3Ib), **P = 0.0015 (25 µM nsP3), ***P = 0.0001 (50 µM nsP3) for USP10 by one-way ANOVA with Dunnett's multiple comparisons test. **(G)** 1.5 µM G3BP1, 1.5 µM caprin 1, and 20 ng/µl total RNA were coincubated in a three-component system and co-condensation was assessed in the presence of increasing concentrations of G3I compounds. The percent inhibition of G3BP1-GFP in vitro phase separation is shown. Tables show the highest (top) and lowest (bottom) values of an individual curve, LogIC50, the slope at the steepest part of the curve (HillSlope), and IC50. Error bars represent mean ± SD, N = 3 replicates per condition. **(H)** 20 µM purified G3BP1 and 100 ng genomic RNA were coincubated in a two-component system and condensation was assessed in the presence of indicated doses of G3Ib or vehicle control. Condensate formation by G3BP1 and RNA was unaffected by the addition of G3Ib. Scale bar, 30 µm. **(I)** Cytotoxicity assay in U2OS cells treated with indicated concentrations of compounds for 24 h; inhibition of growth was measured by monitoring ATP levels read out through a luciferase signal. N = 2, both replicates are plotted. Source data are available for this figure: SourceData F2.

live cell imaging with and without these compounds (Figley et al., 2014). We began by pretreating cells with varying concentrations of compounds for 20 min, followed by the addition of oxidative stress (500 µM NaAsO₂) to induce stress granule formation (Fig. 3 A). In cells pretreated with vehicle (DMSO), stress granule formation began ~6 min after the addition of sodium arsenite,

followed by a period of growth and fusion that extended until ~20 min after the addition of stress (Fig. 3, B and C).

When cells were pretreated with increasing doses of G3Ia or G3Ib, stress granule formation was robustly inhibited in a dose-dependent manner (Fig. 3, B and C; and Videos 1 and 2). G3Ia reduced total stress granule area by ~36% at 5 µM, ~59% at

20 µM, and ∼80% at 50 µM. G3Ib inhibited stress granule formation even more potently (Fig. 3, B and C; and Videos 1 and 2), reducing total stress granule area by ∼50% at 5 µM, ∼85% at 20 µM, and ∼93% at 50 µM. Stress granule formation was not significantly altered by the addition of either of the inactive enantiomers (G3Ia′ or G3Ib′) at any tested dose (Fig. 3, B and C; and Videos 3 and 4). Preincubation with G3Ia or G3Ib inhibited stress granule formation somewhat more effectively when the concentration of sodium arsenite was reduced by half (250 µM NaAsO$_2$), with G3Ia reducing stress granule formation by ∼86% at 50 µM and G3Ib reducing stress granule formation by ∼94% at 50 µM compared with their inactive enantiomers (Fig. S2, A–C). To test whether G3Ia and G3Ib would have comparable effects in other cell lines, we next examined stress granule formation in HeLa cells stably expressing G3BP1-GFP. Similar to our observations in U2OS cells, we found that 50 µM G3Ia and G3Ib inhibited stress granule formation in HeLa cells by ∼72% and ∼82%, respectively, compared with their inactive enantiomers (Fig. S2, D and E).

Because our live-cell assays focused exclusively on G3BP1, it was formally possible that the G3I compounds did not result in inhibition of stress granule assembly but rather prevented G3BP1 from localizing to otherwise intact stress granules. To test this possibility, we used immunofluorescence to examine additional stress granule markers in cells pretreated with 50 µM G3I compounds and then exposed to 250 µM NaAsO$_2$ for 30 min. We found that treatment with G3Ia or G3Ib, but not their inactive enantiomers, resulted in reduced eIF3η and PABPC1 puncta (Fig. S2 F), demonstrating inhibition of stress granule assembly beyond simple extraction or exclusion of G3BP1.

The incorporation of TDP-43 protein into stress granules has led to speculation that the deposition of TDP-43 into stress granules may participate in the pathogenesis of neurological disease (Colombrita et al., 2009). To assess whether G3Ia and G3Ib prevent the accumulation of TDP-43 into stress-dependent granules, we pretreated U2OS cells with 50 µM G3I compounds, exposed cells to 250 µM NaAsO$_2$ for 30 min, and examined TDP-43 by immunofluorescence (Fig. S2 G). We found TDP-43 accumulation in stress granules in cells treated with vehicle, G3Ia′, or G3Ib′, but not in cells treated with G3Ia or G3Ib, demonstrating that G3I compounds prevent mislocalization of TDP-43 protein to stress granules (Fig. S2 H).

For real-time tracking of TDP-43 movement, we employed CRISPR-modified cells where endogenous TDP-43 and G3BP1 were labeled with GFP and mRuby3, respectively. We first pretreated these cells with 50 µM G3Ib compound and then exposed them to 250 µM NaAsO$_2$ for 30 min (Fig. S2 A). Prior to the induction of stress, TDP-43 primarily localized to the nucleus and also formed small cytoplasmic puncta that did not colocalize with G3BP1. Following the addition of 250 µM NaAsO$_2$, we observed that a small proportion of TDP-43 localized to stress granules in cells pretreated with G3Ib′ but not G3Ib (Fig. S2 I; and Videos 5 and 6). Together, these data demonstrate that G3Ia and G3Ib, but not their inactive enantiomers, block the stress-induced accumulation of TDP-43 into stress granules.

We also assessed whether the inhibitory effect of these compounds in dosed cell culture media persisted over an extended time period (Fig. S3 A). To test this, we pretreated U2OS cells with G3I compounds for 24 h prior to exposure to oxidative stress (500 µM NaAsO$_2$ for 30 min). Cells treated with G3Ia showed no substantial impairment of stress granule formation at concentrations of 5 and 20 µM. However, at a concentration of 50 µM, G3Ia inhibited stress granule formation. In contrast, G3Ib nearly completely inhibited stress granule formation at doses of 20 and 50 µM, indicating that G3Ib has a stronger inhibitory effect on stress granule formation compared to G3Ia under these experimental conditions (Fig. S3 B). Treatment with G3Ia′ or G3Ib′ had no discernable effects on stress granule formation (Fig. S3 B).

The composition and the mechanisms of assembly and disassembly of stress granules can vary based on the type of initiating stress (Gwon et al., 2021; Markmiller et al., 2018; Maxwell et al., 2021). With this in mind, we next examined whether G3Ia and G3Ib could prevent stress granule formation following heat stress (Fig. 3 D). We found that preincubation with G3Ia or G3Ib inhibited stress granule formation, with G3Ia reducing stress granule formation by ∼79% at 50 µM and G3Ib reducing stress granule formation by ∼82% at 50 µM (15 min post heat shock) compared with their inactive enantiomers (Fig. 3, E and F).

Taken together, these results demonstrate that G3Ia and G3Ib effectively prevent stress granule growth and fusion in a dose-dependent manner in multiple cell types and in response to different types of cellular stress.

## Treatment with G3Ia and G3Ib rapidly dissolves preformed stress granules

We next tested whether G3Ia and G3Ib could cause the dissolution of preformed stress granules. For these experiments, we first treated U2OS cells stably expressing G3BP1-GFP with sodium arsenite (250 µM) for 30 min to induce stress granule formation. We then added G3Ia, G3Ib, or their inactive enantiomers and monitored the cells using live cell imaging. Remarkably, the addition of 50 µM G3Ia or G3Ib nearly instantaneously reduced stress granule area by ∼74% and ∼84%, respectively, whereas inactive enantiomers had no statistically significant effect on stress granule area (Fig. 4, A and B; and Video 7).

We also examined whether G3Ia and G3Ib could dissolve stress granules generated through heat stress. Here, we exposed cells to 43°C heat stress for 25 min, after which we added 50 µM of G3Ia or G3Ib (Fig. 4 C). Following the addition of G3I compounds, we observed an immediate reduction of ∼54% (G3Ia) and ∼68% (G3Ib) in the stress granule area, whereas inactive enantiomers had no immediate effect on the stress granule area compared with pretreatment (Fig. 4 C and Video 8). After 5 min of incubation with compounds, cells were returned to 37°C, at which point the remaining stress granules began to disassemble in all conditions. As expected, the cells treated with G3Ia or G3Ib completed disassembly faster than cells treated with inactive enantiomers (Fig. 4 C and Video 8).

To confirm that the G3I compounds were resulting in disassembly of stress granules, rather than extracting G3BP1 from otherwise intact stress granules, we used immunofluorescence to examine additional stress granule markers in cells exposed to

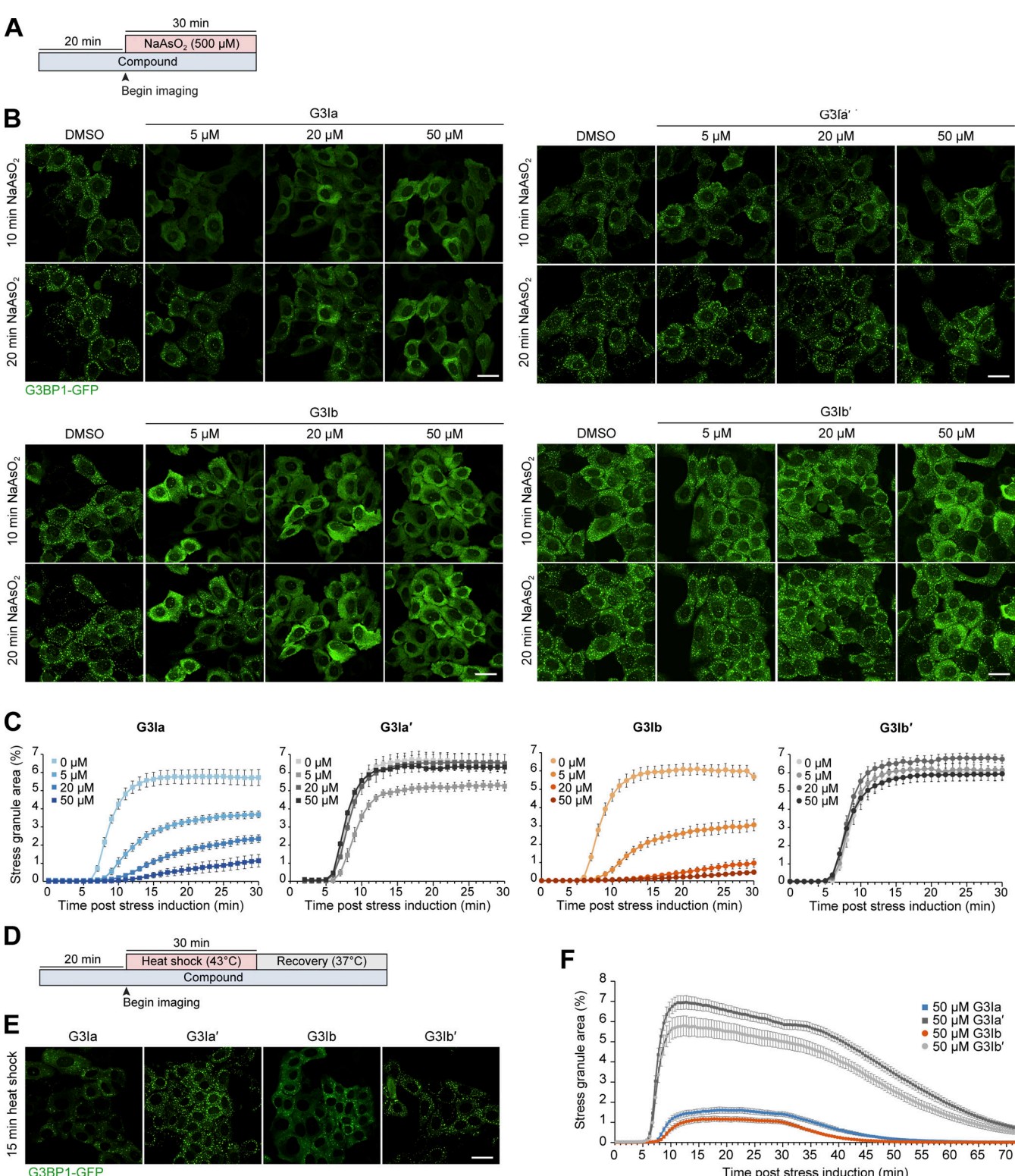

Figure 3. **Preincubation with G3Ia or G3Ib prevents the formation of stress granules in living cells. (A)** Schematic showing the preincubation paradigm used in B and C. Indicated doses of the compound were added to cells for 20 min, followed by exposure to 500 μM NaAsO₂ stress and live cell imaging to monitor stress granule formation. **(B)** Representative images of G3BP1-GFP signal in U2OS cells after 10 or 20 min of oxidative stress. Scale bars, 40 μm. **(C)** Quantification of cells as in B showing the percentage of stress granule area per cell. **(D)** Schematic showing the preincubation paradigm used in E and F. 50 μM of indicated compounds was added to cells for 20 min, followed by exposure to 43°C heat shock for 30 min. Live-cell imaging was used to monitor stress granule formation. **(E)** Representative images of G3BP1-GFP signal in U2OS cells 15 min after heat shock. Scale bar, 40 μm. **(F)** Quantification of cells as in E showing the percentage of stress granule area per cell throughout heat shock (43°C, 30 min) and recovery (37°C, 43 min). Error bars represent mean ± SEM in all graphs.

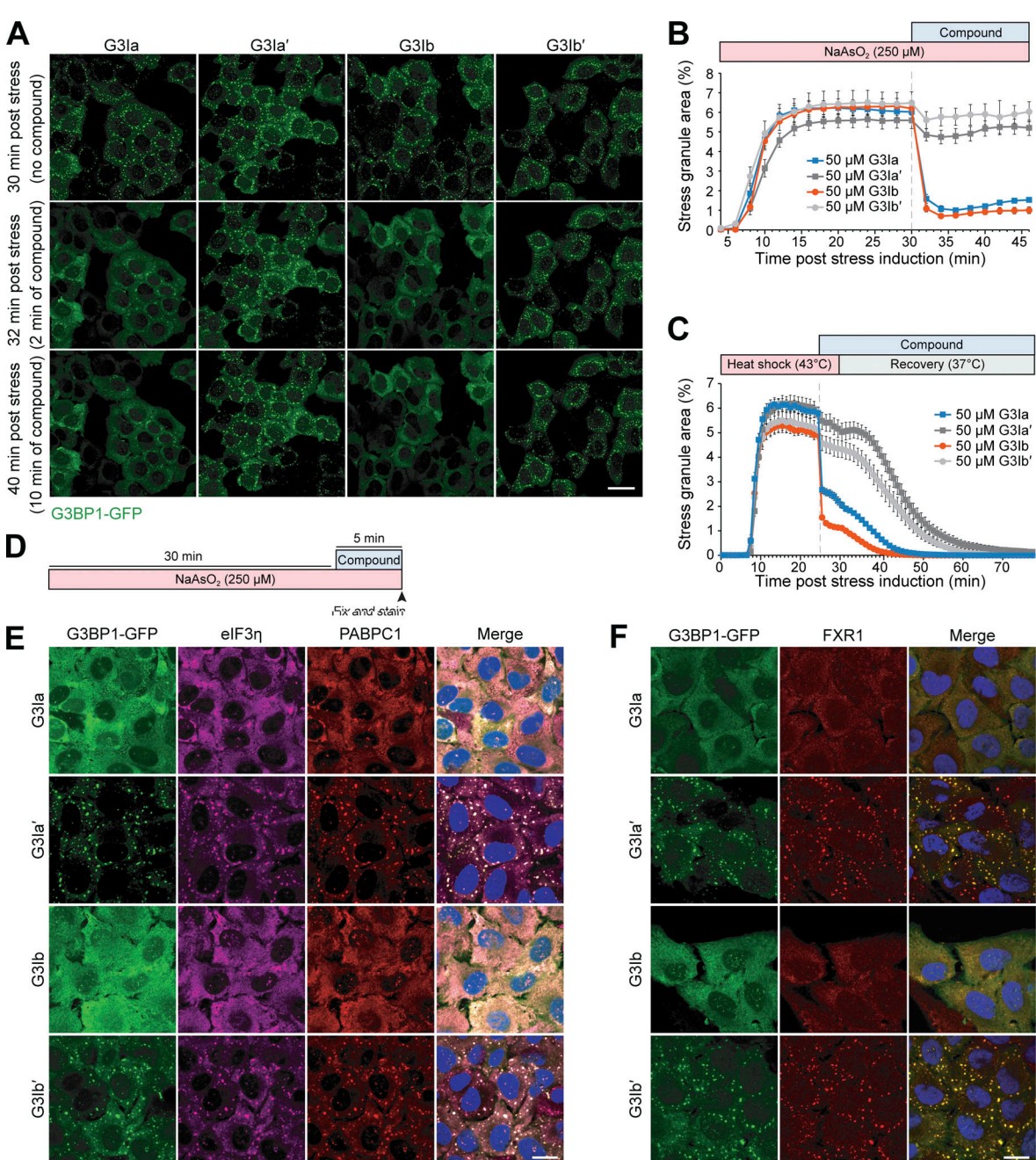

Figure 4. **Treatment with G3Ia and G3Ib rapidly dissolves pre-formed stress granules. (A)** Representative images of G3BP1-GFP signal in U2OS cells following induction of stress by 250 µM NaAsO₂. Images are shown 30 min after induction of stress (immediately before the addition of compound), 32 min after induction of stress (2 min after addition of 50 µM G3I compound), and 40 min after induction of stress (10 min after addition of 50 µM G3I compound). Scale bar, 40 µm. **(B)** Quantification of cells as in A showing the percentage of stress granule area per cell. **(C)** Quantification of the percentage of stress granule area per cell throughout heat shock (43°C, 30 min) and recovery (37°C) from U2OS cells stably expressing G3BP1-GFP. Cells were treated with 50 µM G3I compound 25 min after the induction of heat shock. **(D)** Schematic showing the experimental paradigm used in E and F. U2OS cells stably expressing G3BP1-GFP were exposed to 250 µM NaAsO₂ for 30 min followed by the addition of 50 µM G3I compound. Cells were fixed and stained 5 min after compound was added. **(E and F)** Shown are representative images of immunofluorescence staining of additional stress granule markers (eIF3η, PABPC1, FXR1). Scale bars, 20 µm. Error bars represent mean ± SEM in all graphs.

250 µM NaAsO₂ for 30 min, followed by the addition of 50 µM compound for 5 min (Fig. 4 D). We found that treatment with G3Ia or G3Ib, but not their inactive enantiomers, resulted in reduced eIF3η, PABPC1, and FXR1 puncta, indicating disassembly of stress granules beyond simple extraction of G3BP1 (Fig. 4, E and F).

**Treatment with G3I compounds does not influence the rate of translation in cells**

Because stress granule formation co-occurs with global repression of translation, it is often postulated that stress granules directly contribute to translational repression during stress. Thus, we sought to determine whether G3I compounds

influence the translation rate of cells, either at basal conditions or following stress. To assess the effect of G3I compounds on translation, we pretreated HeLa cells for 15 min with either vehicle or 50 µM G3I compound and then either added sodium arsenite (500 µM) or left the cells unstressed (Fig. S4 A). 15 min later, cells were dosed with puromycin (500 µM) to label newly synthesized proteins (Fig. S4 A). Cells were then lysed, and newly synthesized proteins were visualized by Western blot. As expected, cells treated with sodium arsenite had strongly suppressed translation compared with unstressed cells (Fig. S4 B, lane 2 versus 7). G3I compounds did not alter the translation of newly synthesized protein under basal conditions or in response to arsenite stress (Fig. S4 B). This finding is consistent with evidence that stress granule formation is uncoupled from the loss of global translation in cells (Mateju et al., 2020).

### Treatment with G3Ia prevents the formation of stress granules and dissolves preformed stress granules in human iPSC-derived neurons

Given the evidence that aberrant stress granule condensation may contribute to neurodegenerative disease (Wolozin and Ivanov, 2019), we next tested whether G3I compounds would affect stress granule assembly and/or disassembly in neuronal cells. Here, we used iPSC-derived cortical neurons exposed simultaneously to 500 µM NaAsO$_2$ and either vehicle, 50 µM G3Ia, or 50 µM G3Ia′ for 60 min, at which point we fixed cells and immunostained for G3BP1 (Fig. S5 A). In the absence of stress, ~3% of neurons treated with vehicle had spontaneous stress granules, which increased to ~70% after 60 min of stress (Fig. S5, B and C). Co-incubation with 50 µM G3Ia, but not G3Ia′, blocked both spontaneous and arsenite-induced stress granule formation (Fig. S5, B and C). Next, we examined whether G3I compounds could dissolve preformed stress granules in iPSC-derived cortical neurons that had been exposed to 30 min of 500 µM NaAsO$_2$ stress. Similar to its effects in other cell types, 50 µM G3Ib nearly instantaneously dissolved stress granules in these neuronal cells (Fig. S5 D).

### Treatment with G3Ia or G3Ib dissolves stress granules formed in response to expression of a disease-causing VCP mutant

Stress granules can also form in response to various pathogenic mutations, even in the absence of exogenous stress. For example, stress granules can arise in cells in response to the expression of specific mutant forms of VCP, which in humans leads to the development of multisystem proteinopathy, a condition characterized by degeneration in muscle, bone, and neurons (Pfeffer et al., 2022). One such dominant VCP mutation, VCP A232E, induces stress granules in cell culture and also slows the disassembly of stress granules formed through stress (Buchan et al., 2013). To assess whether G3Ia and G3Ib could alleviate preformed granules formed in response to VCP expression, we transfected GFP-VCP A232E into U2OS cells in which endogenous G3BP1 was labeled with tdTomato via CRISPR. Following expression of GFP-VCP A232E, ~20% of transfected cells contained stress granules (Fig. 5 A). Adding G3Ia to these cells dissolved 83% of these granules, whereas the inactive enantiomer dissolved only 20% of the granules (Fig. 5, A and B).

Similarly, the addition of G3Ib resulted in the dissolution of 69% of these granules, in contrast to a reduction of only 6% by G3Ib′ (Fig. 5, B). These observations indicate that G3Ia and G3Ib can dissolve abnormal stress granules that occur due to the expression of the pathogenic VCP A232E mutant protein.

### Treatment with G3Ia results in the removal of G3BP1 from stress granules formed in response to the expression of disease-causing FUS mutant

Expression of ALS-causing mutations in the RNA-binding protein FUS leads to the redistribution of mutant FUS protein into the cytoplasm, where it accumulates in stress granules (Dormann et al., 2010). To test the ability of our compounds to dissolve FUS-containing granules, we transfected the disease-causing mutant FUS R495X into U2OS cells expressing endogenous G3BP1 tagged with tdTomato as described above and then used automated imaging analysis to examine these cells before and after addition of G3I compounds. Before the addition of the compound, ~40% of cells expressing FUS R495X contained stress granules in which G3BP1 was colocalized with FUS R495X (Fig. 5 C). The addition of G3Ia eliminated 92% of the G3BP1 puncta, whereas the inactive enantiomer eliminated only 6% of G3BP1 puncta. In contrast, neither G3Ia (3%) nor G3Ia′ (4%) was effective in reducing the number of puncta positive for FUS R495X (Fig. 5, C and D). These results suggest that FUS inclusions are more stable than typical stress granules and are not dependent on G3BP1 for their persistence.

Since G3Ia and G3Ib were unable to extract FUS R495X from granules, we next asked whether pretreatment with G3I compounds could block the accumulation of FUS R495X into granules. Since the G3I compounds were found to remain active in preventing stress granule formation when present in cell culture media for 24 h (Fig. S3), we pretreated U2OS cells with G3I compounds or vehicle prior to transfection with FUS R495X. After 24 h, we used automated analysis to identify transfected cells (via florescence intensity) and calculated the average cytoplasmic stress granule area in these transfected cells. As expected, both G3Ia and G3Ib, but not their inactive enantiomers, reduced the area of G3BP1 found in puncta compared with vehicle-treated cells (Fig. 5 E). However, G3Ia and G3Ib failed to block the formation and incorporation of FUS R495X into cytoplasmic granules (Fig. 5 F). From these results, we conclude that inhibition of G3BP1 condensation does not prevent the inclusion of mutant FUS protein into cytoplasmic puncta.

## Discussion

Using the nsP3 peptide as a lead compound (Fig. 1), we designed two novel small molecules, G3Ia and G3Ib, that inhibit co-condensation of G3BP1, caprin 1, and RNA in vitro (Fig. 2) and inhibit the formation of stress granules in cells when added prior to either arsenite or heat stress (Fig. 3). Additionally, G3Ia and G3Ib trigger the disassembly of existing stress granules when introduced to cells following stress granule formation (Fig. 4). The presumed mechanism of action of these compounds is through antagonism of binding between G3BP1/2 and their binding partners within the NTF2L domain, such as caprin 1 and

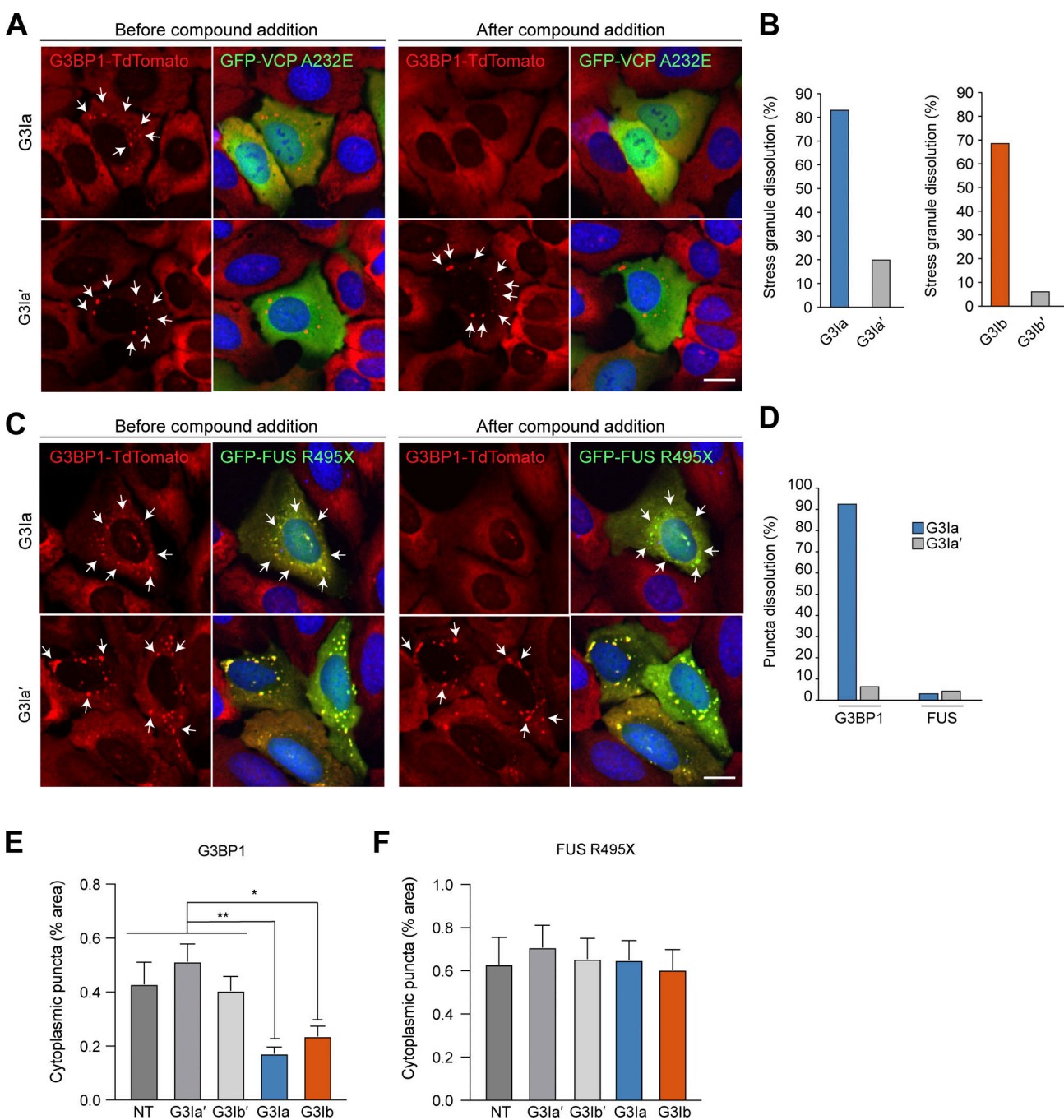

Figure 5. **Treatment with G3I compounds modifies stress granules formed in response to expression of disease-causing mutant proteins. (A)** U2OS cells with TdTomato-tagged endogenous G3BP1 (red) were transfected with VCP A232E (green) for 24 h, then treated with 50 μM G3I compound for 30 min. Shown are representative images of cells before (left) and after (right) addition of G3Ia or G3Ia'. Arrows indicate G3BP1-positive puncta. Scale bar, 10 μm. **(B)** Quantification of cells as in A showing the percentage of stress granule dissolution as assessed by TdTomato imaging. Automated puncta tracking was used for G3Ia; manual blinded cell tracking was used for G3Ib. **(C)** U2OS cells with TdTomato-tagged endogenous G3BP1 (red) were transfected with FUS R495X (green) for 24 h, then treated with 50 μM G3I compound for 30 min. Shown are representative images of cells before (left) and after (right) the addition of G3Ia or G3Ia'. Arrows indicate G3BP1- or FUS R495X-positive puncta. Scale bar, 10 μm. **(D)** Quantification of cells as in C showing the percentage of puncta dissolution for G3BP1-positive and FUS R495X-positive puncta. **(E and F)** U2OS cells with TdTomato-tagged endogenous G3BP1 were pretreated with 50 μM G3I compound or vehicle for 20 min prior to transfection with FUS R495X for 24 h. Using automated analysis pooled from 10 transfected wells per treatment group, the percentage of cytoplasm containing G3BP1 puncta (E) and FUS R495X puncta (F) was quantified in transfected cells. Error bars represent the mean cytoplasmic puncta area per cell ± SEM. *P < 0.1, **P < 0.001 by one-way ANOVA with Dunnett's multiple comparisons test.

USP10 (Fig. 2). The loss of these protein interactions within the NTF2L domain is predicted to limit oligomerization of G3BP1/2 proteins, thus inhibiting the valency necessary for the formation of the large multimeric protein–RNA complexes (Sanders et al., 2020) that comprise stress granules.

The use of G3Ia and G3Ib to target the NTF2L domain of G3BP1/2 has several potential applications. Primarily, these compounds will provide a tool for researchers to manipulate G3BP1/2-dependent stress granule formation and identify specific functions of stress granules in cells. Unlike many other stress granule inhibitors such as cycloheximide (Mollet et al., 2008) that lead to global inhibition of translation and are highly toxic, G3Ia and G3Ib are designed to be specific in their inhibition of protein binding within the NTF2L domain of G3BP1/2. G3Ia and G3Ib impair condensate formation in a highly tractable manner, do not appear to induce cellular toxicity or growth phenotypes (Fig. 2), and are versatile among different cell types. While NTF2L is also the dimerization domain of G3BP1/2 proteins, G3Ia and G3Ib bind to an interaction surface that is on the opposite face of the dimerization surface, thereby leaving dimerization intact (Fig. 1 D). Thus, G3Ia and G3Ib target only the protein–protein interaction domain of G3BP1/2 and specifically block G3BP1/2 condensate formation while leaving the dimerization and RNA-binding capabilities of G3BP1/2 unaltered. This approach also presents a significant advantage over genetic models in which *G3BP1* and *G3BP2* are knocked out, resulting in a complete loss of function of these proteins along with potential alterations in the expression of other stress granule proteins.

Stress granules have been implicated in several disease states, including neurodegeneration and cancer (Protter and Parker, 2016). These compounds could be utilized in disease models to block stress granule formation or potentially dissolve disease-initiated granules, allowing for a better understanding of the roles of these condensates in disease initiation and progression. Consistent with this possibility, we observed that G3Ia and G3Ib can induce the disassembly of aberrant stress granules triggered by the expression of a pathogenic VCP A232E mutation that causes multisystem proteinopathy (Fig. 5). Additionally, G3Ia and G3Ib can be used to probe the immunological role of stress granules, as G3BP1-mediated stress granules are known to form in response to infection with numerous viral RNAs (Jayabalan et al., 2023). Finally, these compounds can be used to define whether the NTF2L domain-mediated interactions of G3BP1/2 play a role in the normal function of cells outside of their ability to drive condensation. Such studies could lead to the discovery of novel stress granule-independent functions of G3BP1, G3BP2, or their interactors in normal cell biology.

In summary, G3Ia and G3Ib are potent inhibitors of binding to the NTF2L domain of G3BP1/2 that are highly specific, easy to synthesize, and highly effective across multiple cell types and stressors, providing a valuable new tool in the study of condensate biology.

## Materials and methods
### Protein purification
All G3BP and caprin 1 proteins were expressed and purified from *E. coli* BL21 (DE3) cells. *E. coli* were grown to $OD_{600}$ of 0.8 and induced with 0.6 mM IPTG at 16°C overnight. Pelleted cells were resuspended in lysis buffer (25 mM Tris, 500 mM NaCl, 1 mM TCEP, 5% glycerol, 20 mM imidazole, pH 7.5, PMSF, and cocktail III [Thermo Fisher Scientific]). After sonication, lysates were pelleted at 30,000 × $g$ at 4°C for 30 min. Supernatants from all His-tagged constructs were purified over Ni Sepharose Fast Flow beads (GE), prewashed with lysis buffer at room temperature, and eluted with 500 mM imidazole (Sigma-Aldrich) in lysis buffer. For TEV cleavage, eluted proteins were incubated with TEV protease at 4°C overnight and then repurified over Ni-FF column. Supernatant from the FLAG-GFP-G3BP1 construct was purified over anti-FLAG resin and then further purified over a Q-HP column using the manufacturer's protocol.

### Surface plasmon resonance (SPR)
Binding affinity for G3BP1 was measured by SPR assay using a Biacore 8K instrument (Cytiva) at 25°C at pH 7.4 (20 mM Tris, 300 mM NaCl, 0.05% Tween-20, 2% DMSO) using neutravidin sensor chips. The binding of solution phase molecules was measured to sensor-bound biotinylated, AVI-tagged versions of the NTF2L domain of G3BP1. G3BP1 was captured on the sensor surface at the beginning of each experiment at a level of ~1,000 RU. A neutravidin-only surface was used as a reference. Binding affinity was measured by equilibrium analysis of double-reference subtracted data performed at seven different analyte concentrations. A 1:1 binding model was used to estimate affinity; the maximum binding response was typically about 90% of the theoretical maximum for a 1:1 interaction given the size of the protein and analytes. Binding at the end of a 90-s association phase was measured; data were only used where equilibrium binding was reached. No regeneration was performed between binding cycles as all tested molecules dissociated completely from the sensor at the end of each cycle. Estimation of binding affinity by kinetic analysis was not performed as kinetic constants were beyond the range that could be reliably measured by the instrumentation.

### Peptide displacement
Peptide displacement assays were performed in 20 μl total volume at 25°C at pH 7.4 in buffer containing 20 mM Tris-HCl, pH 7.5, 100 mM NaCl, 0.01% bovine serum albumin, 1 mM DTT, and 0.005% Tween-20. 10 nM full-length 6x-His-tagged human G3BP1 was added to the reaction mix containing the test compound and 0.5 nM anti-His-Tb antibody (PerkinElmer). Assays were initiated by adding 4 nM FAM-labeled PEG6-USP10 24-mer peptide probe (FAM-PEG6-GALHSPQYIFGDFSPDEFNQFFVT; Peptide 2.0). Assay plates were shaken for 30 s, then centrifuged for 30 s, and then incubated at room temperature for 60 min. Data were read on an EnVision Multilabel plate reader at 495/520 nm. HTRF ratios were calculated as described by PerkinElmer and $IC_{50}$s were calculated using GraphPad Prism 9 software and the [Inhibitor] versus Response—Variable Slope (four parameter) fit equation.

### Crystallography studies and structure analysis
The NTF2L domain of G3BP1 (10 mg/ml) co-crystallized with compound G3Ia at a 3× molar ratio at 4°C in 0.1 M MES pH 6.5

and 15% PEG MME using the sitting drop method. All drops were set up as a 1:1 mixture of protein to reservoir solution. Crystals were cryoprotected with 10–15% glycerol and flash-cooled. Data were collected at the ESRF ID30A1 beamline with a Pilatus3 2M detector in cryostream at 100 K. Data processing and scaling were performed with DIALS (Winter et al., 2018). The diffraction data was phased by molecular replacement using PDB accession no. 5FW5 as the search model (Schulte et al., 2016). The initial model was built by Phaser MR (McCoy et al., 2007) and completed with Coot (Emsley and Cowtan, 2004). Refmac was used to improve density between rounds of manual building (Vagin et al., 2004). Data collection and refinement statistics are shown in Table S1. Coordinates of structures from this study have been deposited in the Protein Data Bank with the accession code 8V1L. Structure analysis and images of binding pockets were made in PyMol.

### LLPS
LLPS assays were performed in 30 µl total volume at 25°C at pH 7.4 in buffer containing 37 mM Tris, pH 7.5, 116 mM NaCl, 0.33% NP-40, 2.6% glycerol, 0.5 mM DTT with 20 ng/µl total cellular RNA from U2OS cells and 1.425 µM G3BP1, 0.075 µM FLAG-GFP-G3BP1, and 1.5 µM caprin 1. Reactions were initiated by adding G3BP1/caprin 1 protein mix to cellular RNA, followed by 3 min of shaking at 450 rpm and then incubation for 70 min in the dark. For condensates formed with G3BP1 and RNA, 20 µM purified G3BP1 and 100 ng of total cellular RNA from HEK293T cells were added to a buffer consisting of 100 mM NaCl and 16.7 mM HEPES (pH 7.5) in the presence of the indicated dose of G3Ib or vehicle control.

### Cell culture and transfection
U2OS (HTB-96), HEK293T (CRL-3216), and HeLa (CCL-2) cells were originally purchased from ATCC and periodically authenticated by short tandem repeat (STR) profiling. Cells were grown in Dulbecco's modified Eagle's medium (DMEM) supplemented with 10% fetal bovine serum (FBS), 1% penicillin/streptomycin, and 1% L-glutamine. Cells were counted using ADAM-CellT (NanoEntek), plated, and transfected using either Lipofectamine 3000 (L3000008; Thermo Fisher Scientific) or Viafect (E4981; Promega) for transient overexpression according to the manufacturer's instructions. Cytotoxicity was assayed using CellTiter-Glo 2.0 reagent (Promega) according to the manufacturer's instructions.

### Preincubation of compounds and live cell imaging
For preincubation of compounds, U2OS or HeLa cells stably expressing G3BP1-GFP or CRISPR-modified cells expressing TDP-43-GFP/G3BP1-GFP were seeded into eight-well Lab-Tek chambered cover glass (Nunc). At least 20 min prior to the experiment, 1 ml FluoroBrite DMEM media (Gibco) supplemented with 10% FBS and 4 mM L-glutamine was combined with 0.1% DMSO or G3I compound (5, 10, or 50 µM). This solution was vortexed for 5 s, added to the cells at 250 µl per well, and left to incubate on the microscope for 20 min. Conditions were maintained at 37°C and 5% $CO_2$ using a Bold Line Cage Incubator (Okolab) and an objective heater (Bioptechs). During this

incubation time, 5 xy fields were stored per condition, with each field having ~20–30 cells within it. After the 20-min incubation, 250 µl of 500 µM or 1 mM (2×) sodium arsenite (Sigma-Aldrich) diluted in the FluoroBrite media with an appropriate amount of G3I compound or DMSO was added to the sample. Imaging began immediately after, with images at each xy field being taken every 1 min. For heat shock experiments, U2OS G3BP1-GFP-expressing cells were seeded into a 35-mm glass bottom dish (MatTek). Prior to the experiment, 1.5 ml FluoroBrite DMEM media (Gibco) supplemented with 10% FBS and 4 mM L-glutamine was combined with 0.1% DMSO or 50 µM G3I compound. The sample was incubated on the environmentally controlled microscope for 20 min. During this incubation time, 10 xy fields were stored per condition. After the 20-min incubation, acquisition began with each xy position being imaged every 30 s. 2 min into the acquisition, the objective heater was ramped to 43°C and maintained this temperature until ramped back down to 37°C at 32 min into the acquisition. All imaging was acquired on a Yokogawa CSU W1 spinning disk attached to a Nikon Ti2 eclipse with a Photometrics Prime 95B camera using Nikon Elements software (v5.20.00 to v5.21.02). Imaging was performed through a Nikon Plan Apo 60× 1.40 NA oil objective with Immersol 518 F (refractive index 1.518; Zeiss), and Perfect Focus 2.0 (Nikon) was engaged for all captures.

### Post-addition of compounds and live cell imaging
For post-addition of compounds, U2OS cells stably expressing G3BP1-GFP were seeded into eight-well Lab-Tek chambered cover glass (Nunc) for sodium arsenite experiments or 35-mm glass-bottom dishes (MatTek) for heat shock experiments. Immediately prior to the experiment, media was replaced with FluoroBrite DMEM media (Gibco) in each well supplemented with 10% FBS and 4 mM L-glutamine; for sodium arsenite experiments, arsenite was added at a final concentration of 250 µM. Cells were then placed on a microscope on which 37°C temperature and 5% $CO_2$ were maintained using a Bold Line Cage Incubator (Okolab) and an objective heater (Bioptechs), and 5–10 xy fields were stored. For sodium arsenite experiments, acquisition began immediately after finding all xy positions, 4 min after the addition of sodium arsenite. 30 min after arsenite was added, 250 µl FluoroBrite DMEM with 250 µM sodium arsenite and 0.1 mM (2×) G3I compound (vortexed for 5 s) was added to the sample. For heat shock experiments, each xy position was imaged every minute. At 2 min into the acquisition, the objective heater was ramped to 43°C. Following this, 250 µM (5×) G3I compound was prepared in FluoroBrite DMEM and vortexed for 15 s. This solution was kept on a 47°C heater near the microscope along with a pipette tip. At 27 min into the acquisition (25 min into heat shock), 250 µl of the prewarmed solution was added to the cells using the prewarmed tip. We note that the solution was heated to 47°C to compensate for the rapid heat loss when adding the solution. At 32 min into the acquisition (30 min heat shock), the heat was ramped down to 37°C. All imaging was acquired on a Yokogawa CSU W1 spinning disk attached to a Nikon Ti2 eclipse with a Photometrics Prime 95B camera using Nikon Elements software (v5.20.00 to v5.21.02). Imaging was performed through a Nikon Plan Apo 60× 1.40 NA

oil objective with Immersol 518 F (refractive index 1.518; Zeiss), and Perfect Focus 2.0 (Nikon) was engaged for all captures.

## Image analysis of preincubation and post-addition of compounds

Automated granule detection and measurement were performed using a combination of ilastik and CellProfiler software. Briefly, ND2 multipoint timelapse files were resaved as image sequences in Fiji with the Bio-Formats plugin so that individual frames of the movies could be treated as individual images for analysis. From there, pixel classification in ilastik was used to segment both stress granules and the total cellular area within the field using the G3BP1-GFP channel via machine learning. These segmentations, along with the original G3BP1-GFP image, were input into CellProfiler, where objects were defined using the masks from ilastik. Using these data points, the measurement of the granules and total cellular area was performed.

## Immunofluorescence in cell lines

U2OS cells or U2OS cells stably expressing G3BP1-GFP were seeded in four-well Lab-Tek II Chambered Coverglass (1055360; Thermo Fisher Scientific). For preincubation experiments, cells were treated with 50 μM G3I compound for 20 min or 24 h, followed by 250 μM NaAsO$_2$ and 50 μM G3I compound for 30 min. For posttreatment, cells were treated with 250 μM NaAsO$_2$ for 30 min, followed by treatment with 250 μM NaAsO$_2$ and 50 μM G3I compound for 5 min. Cells were then washed twice with PBS and fixed with 4% PFA in PBS for 10 min. Cells were then washed three times with PBS (5 min between each wash), permeabilized with 0.5% Triton X-100 in PBS for 10 min, and washed twice with PBS. Next, cells were blocked for 1 h with 3% BSA in PBS at room temperature, followed by incubation with primary antibodies diluted in 3% BSA overnight at 4°C. Cells were then washed three times with PBS (5 min between each wash) and incubated with secondary antibodies and Hoechst diluted in 3% BSA at room temperature for 1 h. Cells were then washed three times with PBS (5 min between each wash) and imaged. Primary antibodies were as follows: G3BP1 (Cat# 611127, RRID:AB_398438; BD Biosciences) at 1:200; TDP-43 (Cat# 12892-1-AP, RRID:AB_2200505; Proteintech) at 1:50; eIF3η (Cat# sc-137214, RRID:AB_2277705; Santa Cruz Biotechnology) at 1:100 dilution; PABP (Cat# ab21060, RRID:AB_777008; Abcam) at 1:1,000 dilution; and FXR1 (Cat# ab51970, RRID:AB_880113; Abcam) at 1:100 dilution. Secondary antibodies were as follows: donkey anti-mouse Alexa Fluor 647 (Cat# A-31571, RRID: AB_162542; Thermo Fisher Scientific), donkey anti-rabbit Alexa Fluor 555 (Cat# A-31572 [also A31572], RRID:AB_162543; Thermo Fisher Scientific), and donkey anti-goat Alexa Fluor 555 (Cat# A-21432 [also A21432], RRID:AB_2535853; Thermo Fisher Scientific). Nuclei were stained with Hoechst 33342 (40046; Biotium) at 1:10,000 dilution. All imaging was acquired on a Yokogawa CSU W1 spinning disk attached to a Nikon Ti2 eclipse with a Photometrics Prime 95B camera using Nikon Elements software (versions 5.20.00 to 5.21.02). Imaging was performed through a Nikon Plan Apo 60× 1.40 NA oil objective with Immersol 518 F (refractive index 1.518; Zeiss), and Perfect Focus 2.0 (Nikon) was engaged for all captures.

## Imaging of mutation-induced granules and analysis

U2OS cells with knock-in of G3BP1-tdTomato (Yang et al., 2020) were seeded into 96-well plates (3904; Corning). Either GFP-VCP-A232E or GFP-FUS-R495X were transfected into cells using ViaFect transfection reagent (Promega) 24 h prior to the start of the experiment. For samples to be treated with compounds G3Ia or G3Ia′, Hoechst (1:5,000; Biotium) was added and incubated for 30 min prior to the start of the experiment, washed once with PBS, and the media was changed to 100 μl FluoroBrite DMEM media (Gibco) upon starting the experiment. Imaging of plates was performed on a Cytation C10 spinning disk confocal (BioTek) with a Hamamatsu Orca-Flash 4.0 camera using Gen5 software (version 3.11). Imaging was performed through an Olympus Plan Apo 40× 0.6 NA dry objective with an adjustable collar set to 0.5 μm thickness. Laser autofocus prior to imaging the tdTomato channel was utilized for each image. The temperature was maintained at 37°C on the instrument and the cells were supplied with 5% CO$_2$. A 7 × 7 tilescan was taken at the center of each well, capturing the tdTomato and GFP channels (and Hoechst when present). After completing the tilescan, 100 μl FluoroBrite with 100 μM (2×) G3I compound (vortexed for 5 s) was added to each well and incubated for 20 min before starting another round of imaging in the same locations as the first round. Samples treated with G3Ib or G3Ib′ were analyzed manually, scoring each cell with spontaneous granules for the presence or absence of granules following treatment with the compound. For experiments in which G3I compounds were pretreated prior to transfection, 50 μM of indicated compound or vehicle control were added to U2OS cells 20 min prior to the addition of the transfection reagent. After 24 h of transfection, the cells were washed with PBS, fixed in 4% formaldehyde, and incubated with Hoechst (1:5,000; Biotium) in PBS for 30 min prior to the start of the experiment, washed once with PBS, and at room temperature imaging on the Cytation C10 spinning disk confocal was performed as described above.

Samples treated with G3Ia or G3Ia′ were analyzed using an automated pipeline. Segmentation was performed in both ilastik and Cellpose 2.0. Nuclear segmentation using the Hoechst channel and granule segmentation in both GFP and tdTomato channels was performed via pixel classification in ilastik using small cropped portions of the tilescan as the training dataset for each. Individual cell segmentation was performed in Cellpose 2.0 by inputting a merged RGB of the Hoechst and G3BP1-tdTomato channels. The Cellpose output was then eroded by one pixel and set as a binary mask in Fiji. The original three raw channels, three output masks from ilastik, and the one mask generated from Cellpose were all inputted into CellProfiler to generate per-cell measurements for granules as defined by GFP or tdTomato.

## Coimmunoprecipitation

U2OS cells expressing G3BP1-GFP were grown in 3-cm dishes to 100% confluency, at which point cells were collected and pelleted at 400 × g for 5 min. Pellets were either stored at –80°C or processed the same day by lysing with 200 μl in vivo lysis buffer (150 mM NaCl, 50 mM Tris [pH 7.5], 1 mM EDTA, 0.5% NP-40, and 10% glycerol, supplemented with 1 protease inhibitor tablet

[11836170001; Roche]). Lysates were centrifuged at 13,000 × *g* at 4°C for 15 min, after which supernatants were transferred to a new tube and diluted with 200 µl lysis buffer containing indicated concentrations of G3I compounds. The mixture was then incubated with GFP-Trap (ChromoTek GFP-Trap Magnetic Agarose; Proteintech; gtma) for 2 h at 4°C. Beads were washed three times with lysis buffer, then incubated with 1× SDS sample buffer supplemented with 10% 2-mercaptoethanol, and boiled for 5 min at 95°C. Lysates were subjected to SDS-PAGE and then immunoblotted.

## Western blotting

Samples were separated by SDS-PAGE and transferred to membranes. Western blots were probed using the following primary antibodies: anti-caprin 1 (Cat# 15112-1-AP, RRID: AB_2070016; Proteintech) or (Cat# 66352-1-Ig, RRID:AB_2881732; Proteintech) both at 1:1,000; anti-USP10 (Cat# A300-900A, RRID: AB_625312; Bethyl) at 1:1,000; anti-GAPDH (Cat# sc-32233, RRID: AB_627679; Santa Cruz Biotechnology) at 1:1,000; anti-GFP (Cat# 11814460001, RRID:AB_390913; Roche) at 1:1,000; anti-puromycin (Cat# MABE343, RRID:AB_2566826; Millipore) at 1:1,000; and anti-actin (Cat# sc-1616, RRID:AB_630836; Santa Cruz Biotechnology) at 1:1,000 at 4°C overnight. Membranes were washed three times with PBST (0.1% Tween) and further incubated with dye-labeled secondary antibodies: donkey anti-mouse IRDye 800CW, (Cat# 926-32212, RRID:AB_621847; LI-COR Biosciences); and donkey anti-rabbit IRDye 680RD, (Cat# 926-68073, RRID:AB_10954442; LI-COR Biosciences). Membranes were visualized and quantified with an Odyssey Fc imaging system (LI-COR) and normalized to the amount of G3BP1 immunoprecipitated.

## iPSC neuron differentiation

i3N iPSC clones were gifts from Michael E. Ward (National Institutes of Health, Bethesda, MD, USA) and were used to generate a G3BP1-mNeongreen knock-in iPSC line. iPS neurons were differentiated with a two-step protocol (predifferentiation and maturation) as previously described (Wang et al., 2017). For predifferentiation, iPSCs were incubated with 1 µg/ml doxycycline hyclate (Sigma-Aldrich) for 3 days at a density of $1.2 \times 10^6$ cells/well in six-well dishes coated with Matrigel in knockout DMEM (KO-DMEM)/F12 medium (Thermo Fisher Scientific) containing N2 supplement (Thermo Fisher Scientific), non-essential amino acids (NEAA; Thermo Fisher Scientific), GlutaMAX Supplement (Thermo Fisher Scientific), and Y-27632 (STEMCELL Technologies). The medium was changed daily and Y-27632 was removed from day 2. For maturation, predifferentiated precursor cells were dissociated, counted, and subplated at $25 \times 10^4$ cells/ml on dishes coated with 50 µg/ml poly-L-ornithine in BrainPhys neuronal medium (STEMCELL Technologies) containing N2 (Thermo Fisher Scientific), B-27, 20 ng/ml BDNF (PeproTech), 20 ng/ml GDNF (PeproTech), 500 µg/ml dibutyryl cyclic-AMP (Sigma-Aldrich), 200 nM L-ascorbic acid (Sigma-Aldrich), 1 µg/ml natural mouse laminin (Thermo Fisher Scientific), 1 µM Ara-C (Sigma-Aldrich), and 1 µg/ml doxycycline hyclate. Half-medium was changed every other day.

## Immunocytochemistry in iPSC-derived neurons

DIV 21 human iPSC cortical neurons were coincubated with compounds (50 µM or vehicle) and 500 µM NaAsO$_2$ for 1 h, then immunostained as described previously (Ippolito and Eroglu, 2010) with modifications. In brief, after fixation with 4% PFA, neurons were blocked with PBS containing 5% normal goat serum and 0.1% Triton X-100 for 1 h. Primary antibodies were diluted with TBST containing 1% normal goat serum and 0.1% Triton X-100. After overnight incubation of primary antibodies at 4°C, neurons were washed three times with TBST, two times with PBS, and incubated with Alexa Fluor 488, Alexa Fluor 555, and Alexa Fluor 633-conjugated secondary antibodies (Invitrogen) at a dilution of 1:1,000 for 2 h at room temperature. Images were captured using a Cytation microscope.

## Assembly and disassembly of stress granules using time-lapse live-cell microscopy in iPSC-derived neurons

DIV 21 human iPSC cortical neurons were incubated with 500 µM NaAsO$_2$ for 30 min, after which 50 µM compound or vehicle was added. Live-cell imaging was performed using a Yokogawa CSU W1 spinning disk. A Yokogawa CSU W1 spinning disk attached to a Nikon Ti2 eclipse with a Photometrics Prime 95B camera using Nikon Elements software was used in time-lapse live-cell imaging. Imaging was taken using a 60× Plan Apo 1.4 NA oil objective and Perfect Focus 2.0 (Nikon) engaged for the duration of the capture. During imaging, cells were maintained at 37°C and supplied with 5% CO$_2$ using a Bold Line Cage Incubator (Okolab) and an objective heater (Bioptechs). To monitor the assembly and disassembly of stress granules, multipoint images over five xy fields for each condition per one replicate were taken with the 488-nm laser. Images were taken at each xy position every 1 min.

## Statistical analysis

P values were determined using one-way ANOVA with Dunnett's multiple comparisons test and error bars are either standard deviation or standard error of the mean as indicated within the figure legend. Data distribution was assumed to be normal, but this was not formally tested.

## Online supplemental material

Fig. S1 shows the schematic for how the G3I compounds were identified. Fig. S2 shows the effect of preincubation of G3I compounds following exposure to 250 µM NaAsO$_2$ in both U2OS and in HeLa cells, and how the G3I compounds influence the localization of other stress granule proteins. Fig. S3 shows the effect of G3I compounds following pretreatment for 24 h prior to exposure to 500 µM NaAsO$_2$. Fig. S4 shows that G3I compounds do not influence global translation rates. Fig. S5 shows that G3I compounds prevent and dissolve stress granules in human-IPSC neurons. Videos 1, 2, 3, and 4 show live cell imaging of G3BP1 in U2OS cells exposed to 500 µM NaAsO$_2$ pretreated with G3I compounds: G3Ia (Video 1), G3Ib (Video 2), G3Ia′ (Video 3), and G3Ib′ (Video 4). Videos 5 and 6 show live cell imaging of G3BP1 and TDP-43 in U2OS cells pretreated with G3Ib′ (Video 5) or G3Ib (Video 6). Video 7 shows live cell imaging of G3BP1 in U2OS cells exposed to 250 µM NaAsO$_2$ followed by treatment with G3I

compounds. Video 8 shows live cell imaging of G3BP1 in U2OS cells exposed to heat shock followed by treatment with G3I compounds. Table S1 shows crystallographic statistics for G3BP1 NTF2L domain with G3Ia.

## Acknowledgments

We thank Natalia Nedelsky for editorial assistance. We thank Michael Hughes for his help with deposition crystal structures in the RCSB Protein Data Bank. We thank Eamon Comer for medicinal chemistry insights and discussion.

This work was supported by the Howard Hughes Medical Institute and National Institutes of Health grant R35NS097974 to J.P. Taylor.

Author contributions: J. Hixon and R. Lemieux designed compounds. B.D. Freibaum, J. Messing, H. Nakamura, U. Yurtsever, J. Wu, J. Hixon, J. Duffner, W. Huynh, K. Wong, M. White, and C. Lee designed and performed experiments. B.D. Freibaum, J. Messing, H. Nakamura, U. Yurtsever, J. Wu, H.J. Kim, R. Lemieux, J. Duffner, W. Huynh, K. Wong, M. White, and C. Lee analyzed data. B.D. Freibaum, H.J. Kim, R. Meyers, R. Parker, and J.P. Taylor drafted and revised the manuscript. R. Meyers and J.P. Taylor supervised the overall study.

Disclosures: All authors have completed and submitted the ICMJE Form for Disclosure of Potential Conflicts of Interest. J. Hixon reported "other" from Faze Medicines outside the submitted work and "Employee and shareholder of Faze Medicines" at the time work was done. W. Huynh reported personal fees from Faze Medicines during the conduct of the study and "I was an employee and stockholder of Faze Medicines" at the time that the experiments were done. K. Wong reported "Employee of Faze Medicines." No other disclosures were reported.

Submitted: 15 August 2023

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

Figure S1. **Identification of lead compounds that interact with the nsP3 binding pocket of the NTF2L domain of G3BP1.** Schematic showing the lead optimization and compound discovery process that led to the identification of G3Ia and G3Ib as lead compounds.

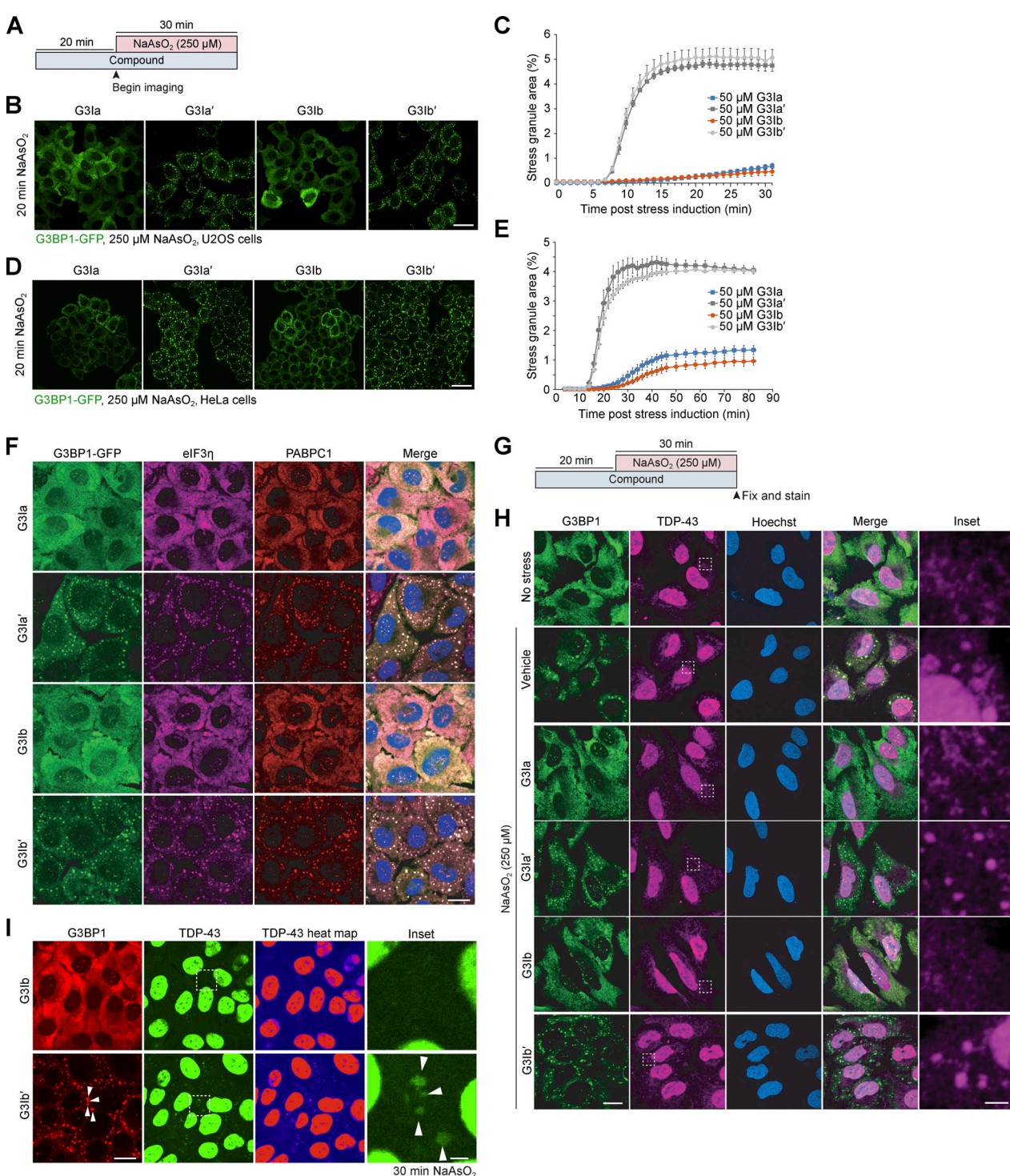

Figure S2. **Preincubation with G3Ia or G3Ib prevents the formation of stress granules and inhibits the accumulation of TDP-43 into stress granules.**
**(A)** Schematic showing the preincubation paradigm used in B–E and I. 50 µM of indicated compounds was added to cells for 20 min, followed by exposure to 250 µM NaAsO2 stress and live cell imaging to monitor stress granule formation. **(B)** Representative images of G3BP1-GFP signal in U2OS cells after 20 min 250 µM NaAsO2. Scale bar, 40 µm. **(C)** Quantification of cells as in B showing the percentage of stress granule area per cell. **(D)** Representative images of G3BP1-GFP signal in HeLa cells after 20 min of 250 µM NaAsO2. Scale bar, 40 µm. **(E)** Quantification of cells in D showing the percentage of stress granule area per cell. **(F)** Representative images of immunofluorescent staining of additional stress granule markers (eIF3η, PABPC1) in cells pre-treated with 50 µM G3I compounds and then exposed to 250 µM NaAsO2 for 30 min. Scale bar, 20 µm. **(G)** Schematic showing the preincubation paradigm used in H. 50 µM of indicated compounds was added to U2OS cells for 20 min, followed by exposure to 250 µM NaAsO2 stress, followed by fixation and immunofluorescence for G3BP1 and TDP-43. Vehicle and unstressed cells were used as controls. **(H)** Representative images of immunofluorescence in U2OS cells after 30 min 250 µM NaAsO2. Scale bars, 20 and 3 µm (inset). **(I)** Representative images of mRuby3-G3BP1 and GFP-TDP-43 signal in U2OS cells after pretreatment with compound followed by 30 min 250 µM NaAsO2. Arrowheads indicate puncta positive for G3BP1 and TDP-43; scale bars, 20 and 3 µm (inset). Error bars represent mean ± SEM in all graphs.

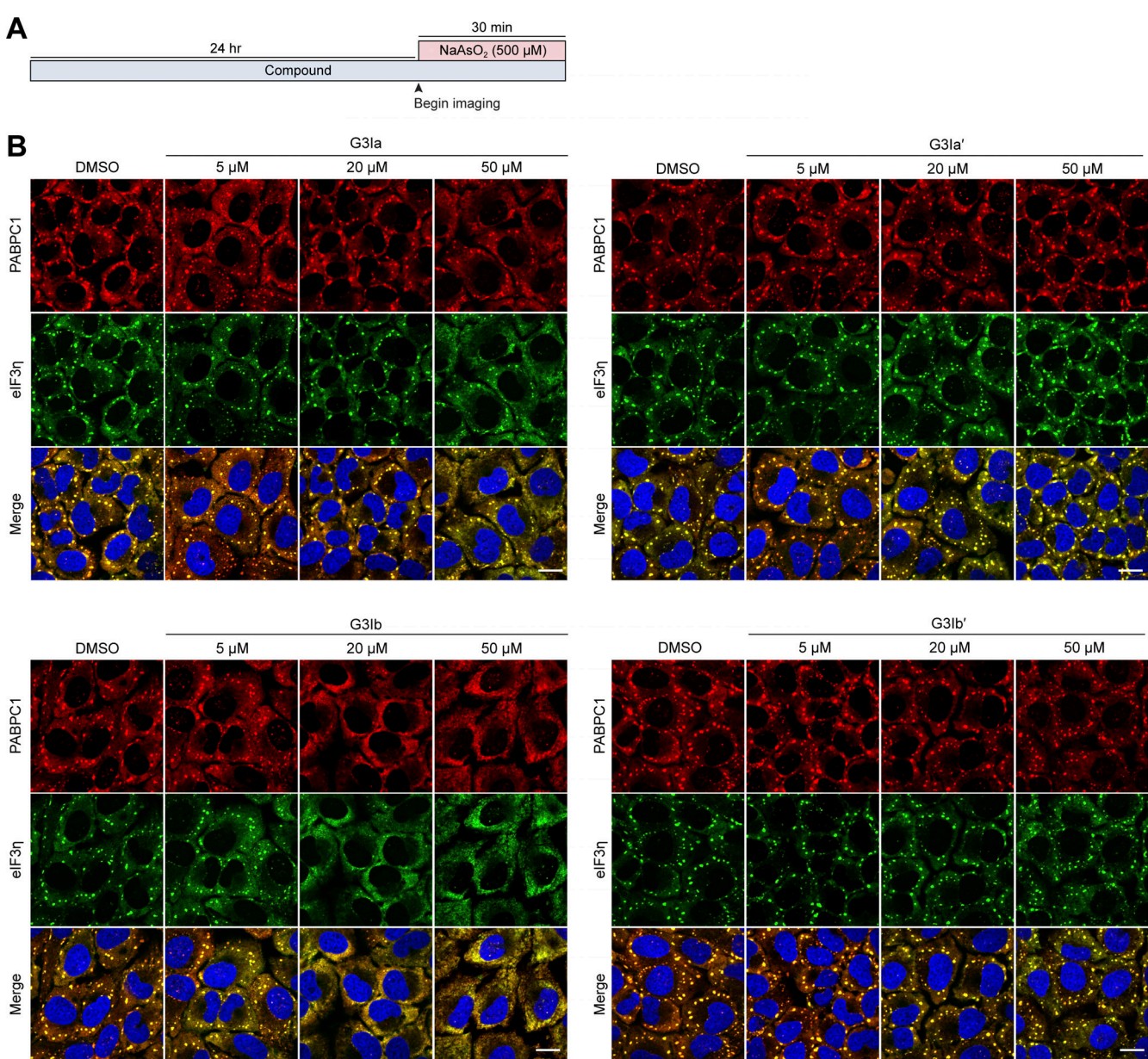

Figure S3. **G3Ia and G3Ib continue to inhibit the formation of stress granules after 24 h of exposure to compounds. (A)** Schematic showing the preincubation paradigm used in B. Indicated doses of the compound were added to cells for 24 h, followed by exposure to 500 μM NaAsO$_2$ stress for 30 min to monitor stress granule formation. **(B)** Shown are representative images of immunofluorescence staining of additional stress granule markers (eIF3η, PABPC1). Scale bars, 20 μm.

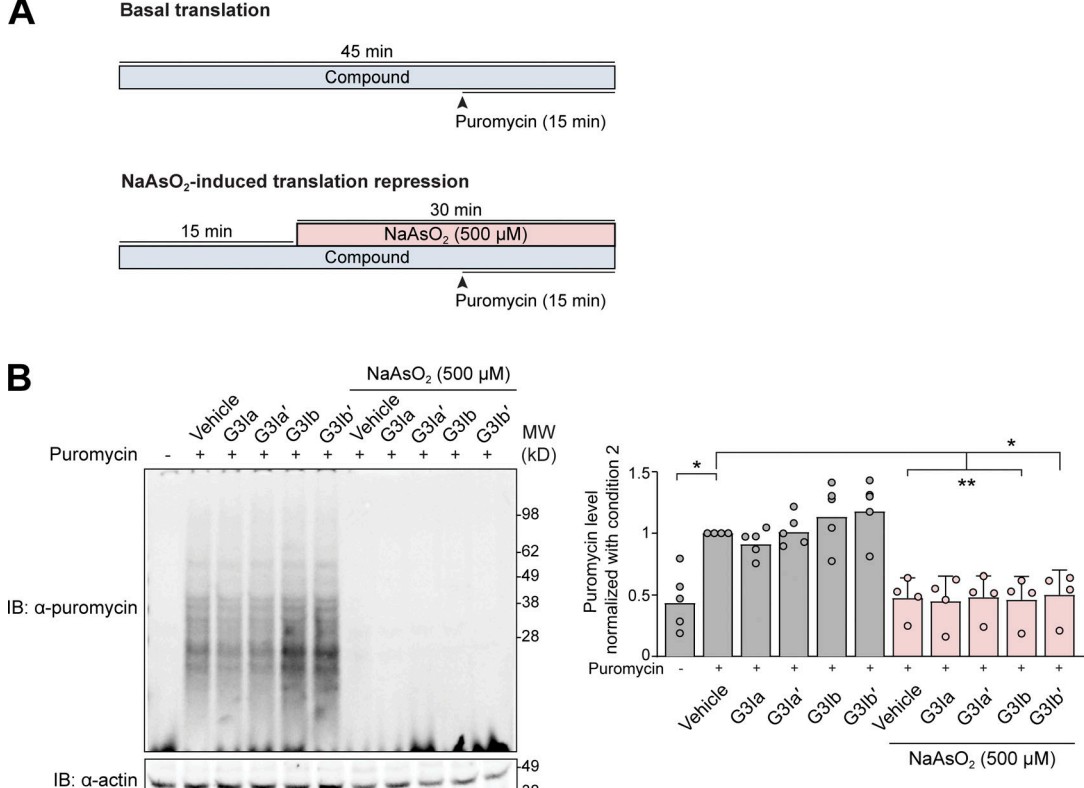

Figure S4. **G3I compounds do not alter translation under basal conditions or following sodium arsenite stress. (A)** Schematic showing the pre-incubation paradigm used for pretreatment with 50 μM of indicated G3I compound or vehicle control. Indicated doses of the compound were added to HeLa cells for 15 min, followed by exposure to 500 μM NaAsO₂ stress for 30 min. Puromycin (500 μM) was added to the media 15 min following the addition of NaAsO₂. Unstressed cells were used to quantify the basal translation rate. **(B)** Cells were collected and lysed with RIPA buffer followed by SDS-PAGE. Newly synthesized transcripts were visualized by Western blot using an antibody targeting puromycin. Actin was used a loading control. Densitometry from $n = 4$ blots was used to generate a graph representing puromycin labeling of newly synthesized proteins. Error bars represent mean ± SD. *$P < 0.1$ and **$P = 0.05$ by one-way ANOVA with Dunnett's multiple comparisons test. Source data are available for this figure: SourceData FS4.

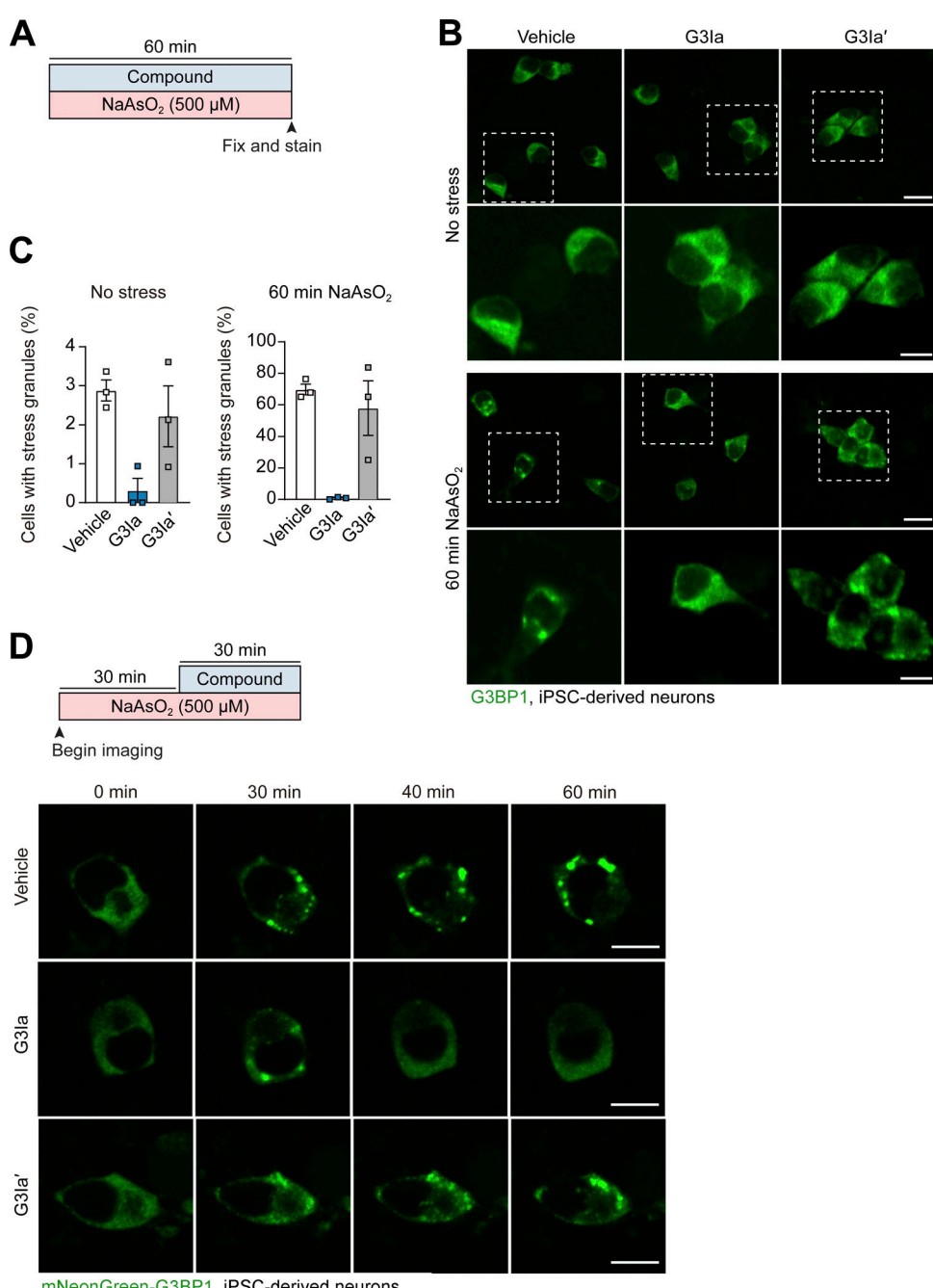

Figure S5.  **Treatment with G3Ia prevents the formation of stress granules and dissolves pre-formed stress granules in human iPSC-derived neurons.** **(A)** Schematic showing the paradigm used in B and C. iPSC-derived cortical neurons were treated with 50 μM G3Ia or G3Ia' in the presence or absence of 500 μM NaAsO$_2$ for 60 min followed by fixation and imaging. **(B)** Representative images showing stress granules via staining of G3BP1 in cortical neurons in the presence or absence of G3I compounds under baseline or stressed conditions. Scale bars, 20 and 5 μm (inset). **(C)** Quantification of cells as in B showing the percentage of cells with stress granules under each condition. Error bars represent mean ± SEM. **(D)** Schematic showing G3BP1-mNeonGreen iPSC-derived neurons exposed to 500 μM NaAsO$_2$ for 30 min, at which point indicated 50 μM G3I compounds were added. Shown are representative images showing live cell imaging of stress granules via mNeonGreen-tagged G3BP1. Scale bars, 10 μm.

Video 1.  **G3Ia inhibits sodium arsenite-induced stress granule formation in a dose-dependent manner.** Live cell imaging of U2OS G3BP1-GFP-expressing cells following preincubation with DMSO, 5, 20, or 50 μM G3Ia (left to right) followed by exposure to 500 μM NaAsO$_2$ for 30 min. Playback speed: 5 frames per second.

Video 2.   **G3Ib inhibits sodium arsenite-induced stress granule formation in a dose-dependent manner.** Live cell imaging of U2OS G3BP1-GFP-expressing cells following preincubation with DMSO, 5, 20, or 50 µM G3Ib (left to right) followed by exposure to 500 µM NaAsO$_2$ for 30 min. Playback speed: 5 frames per second.

Video 3.   **An inactive enantiomer, G3Ia', does not affect sodium arsenite–induced stress granule formation.** Live cell imaging of U2OS G3BP1-GFP-expressing cells following preincubation with DMSO, 5, 20, or 50 µM G3Ia' (left to right) followed by exposure to 500 µM NaAsO$_2$ for 30 min. Playback speed: 5 frames per second.

Video 4.   **An inactive enantiomer, G3Ib', does not affect sodium arsenite–induced stress granule formation.** Live cell imaging of U2OS G3BP1-GFP-expressing cells following preincubation with DMSO, 5, 20, or 50 µM G3Ib' (left to right) followed by exposure to 500 µM NaAsO$_2$ for 30 min. Playback speed: 5 frames per second.

Video 5.   **Following pre-treatment with G3Ib', an inactive enantiomer, TDP-43 localizes to stress granules following sodium arsenite treatment.** Live cell imaging of U2OS cells where TDP-43 (green) and G3BP1 (red) were visualized with GFP and mRuby3 following preincubation with 50 µM G3Ib' followed by exposure to 250 µM NaAsO$_2$ for 30 min. Playback speed: 5 frames per second.

Video 6.   **G3Ib prevents the localization of TDP-43 to stress granules following sodium arsenite treatment.** Live cell imaging of U2OS cells where TDP-43 (green) and G3BP1 (red) were visualized with GFP and mRuby3 following preincubation with 50 µM G3Ib followed by exposure to 250 µM NaAsO$_2$ for 30 min. Playback speed: 5 frames per second.

Video 7.   **G3Ia and G3Ib, but not their inactive enantiomers, dissolve stress granules induced by sodium arsenite.** Live cell imaging of U2OS G3BP1-GFP-expressing cells following exposure to 250 µM NaAsO$_2$ for 30 min followed by treatment with 50 µM G3Ia, G3Ia', G3Ib, or G3Ib' (left to right). Playback speed: 3 frames per second.

Video 8.   **G3Ia and G3Ib, but not their inactive enantiomers, dissolve stress granules induced by heat stress.** Live cell imaging of U2OS G3BP1-GFP-expressing cells following exposure to a 43°C heat shock for 30 min followed by treatment with 50 µM G3Ia, G3Ia', G3Ib, or G3Ib' (left to right) 25 min after exposure to heat shock. Playback speed: 5 frames per second.

**Provided online is Table S1. Table S1 shows crystallographic statistics for G3BP1 NTF2L domain with G3Ia.**

