## [Peer Review File · The Journal of Cell Biology]

Identification of small molecule inhibitors of G3BP-driven stress granule formation

Brian Freibaum, James Messing, Haruko Nakamura, Ugur Yurtsever, Jinjun Wu, Hong Joo Kim, Jeffrey Hixon, Rene Lemieux, Jay Duffner, Walter Huynh, Kathy Wong, Michael White, Christina Lee, Rachel Meyers, Roy Parker, and Joseph Taylor

Corresponding Author(s): Joseph Taylor, St. Jude Children's Research Hospital

Review Timeline:

Submission Date:	2023-08-15
Editorial Decision:	2023-09-18
Revision Received:	2023-11-28
Editorial Decision:	2023-12-11
Revision Received:	2023-12-18

Monitoring Editor: Karla Neugebauer

Scientific Editor: Andrea Marat

Transaction Report:

DOI: <https://doi.org/10.1083/jcb.202308083>

September 18, 2023

Re: JCB manuscript #202308083

Dr. Joseph P Taylor
St. Jude Children's Research Hospital
Cell and Molecular Biology
262 Danny Thomas Place
Memphis, TN 38105

Dear Dr. Taylor,

Thank you for submitting your manuscript entitled "Identification of small molecule inhibitors of G3BP-driven stress granule formation". The manuscript was assessed by expert reviewers, whose comments are appended to this letter. We invite you to submit a revision if you can address the reviewers' key concerns, as outlined here.

As you will see, the reviewers are very enthusiastic about the potential utility of the described small molecule inhibitors. They have provided constructive suggestions, which we hope you agree will further strengthen your study. In particular, we request that you address the toxicity concerns and provide better data on the time course effects as requested by reviewer 2. Providing more detailed insight into the FUS inclusions as suggested by reviewer 1 point 2 should also be attempted. Otherwise, we expect you to address all of the remaining minor reviewer comments in your revised manuscript.

GENERAL GUIDELINES:

Text limits: Character count for an Tools is < 40,000, not including spaces. Count includes title page, abstract, introduction, results, discussion, and acknowledgments. Count does not include materials and methods, figure legends, references, tables, or supplemental legends.

Figures: Toolss may have up to 10 main text figures. Figures must be prepared according to the policies outlined in our Instructions to Authors, under Data Presentation, <https://jcb.rupress.org/site/misc/ifora.xhtml>. All figures in accepted manuscripts will be screened prior to publication.

Supplemental information: There are strict limits on the allowable amount of supplemental data. Toolss may have up to 5 supplemental figures. Up to 10 supplemental videos or flash animations are allowed. A summary of all supplemental material should appear at the end of the Materials and methods section.

Please note that JCB now requires authors to submit Source Data used to generate figures containing gels and Western blots with all revised manuscripts. This Source Data consists of fully uncropped and unprocessed images for each gel/blot displayed in the main and supplemental figures. Since your paper includes cropped gel and/or blot images, please be sure to provide one Source Data file for each figure that contains gels and/or blots along with your revised manuscript files. File names for Source Data figures should be alphanumeric without any spaces or special characters (i.e., SourceDataF#, where F# refers to the associated main figure number or SourceDataFS# for those associated with Supplementary figures). The lanes of the gels/blots should be labeled as they are in the associated figure, the place where cropping was applied should be marked (with a box), and molecular weight/size standards should be labeled wherever possible. Source Data files will be made available to reviewers during evaluation of revised manuscripts and, if your paper is eventually published in JCB, the files will be directly linked to specific figures in the published article.

The typical timeframe for revisions is three to four months. While most universities and institutes have reopened labs and allowed researchers to begin working at nearly pre-pandemic levels, we at JCB realize that the lingering effects of the COVID-19 pandemic may still be impacting some aspects of your work, including the acquisition of equipment and reagents. Therefore,

if you anticipate any difficulties in meeting this aforementioned revision time limit, please contact us and we can work with you to find an appropriate time frame for resubmission. Please note that papers are generally considered through only one revision cycle, so any revised manuscript will likely be either accepted or rejected.

Thank you for this interesting contribution to Journal of Cell Biology. You can contact us at the journal office with any questions, cellbio@rockefeller.edu or call (212) 327-8588.

Sincerely,

Karla Neugebauer, PhD
Monitoring Editor

Andrea L. Marat, PhD
Senior Scientific Editor

Journal of Cell Biology

Reviewer #1 (Comments to the Authors (Required)):

Biomolecular condensation is an emerging principle that underlies many cellular processes. Stress granules are bimolecular condensates that form in the response to cellular stresses and have been connected to the pathogenesis of several diseases, including cancer and neurodegeneration.

The authors have previously discovered that two proteins, G3BP1 and G3BP2, are necessary and sufficient for stress granule formation. In this new paper, they rationally design small molecule G3BP1/2 inhibitors and show that they have potent activity against stress granules formed under diverse conditions. They also show that their compounds can either prevent the formation of stress granules or dissolve pre-existing ones and can do so in neurons. The compounds (but not their inactive enantiomers) can also block stress granule formation induced by mutant neurodegenerative disease proteins VCP and FUS. Importantly, these compounds exhibit little to no cellular toxicity.

This is a very exciting paper and the compounds presented here will be of immediate and broad interest to the cell biology field. The paper is very well written and the data are compelling. I recommend this paper be published in the JCB. I have some comments and suggestions for the authors to consider.

1) In this U2OS cell model, inducing stress granules using arsenite has been shown to also cause mislocalization of another ALS protein, TDP-43, from the nucleus to the cytoplasm. Does treatment with G3BP1/2 inhibitors reduce TDP-43 mislocalization to the cytoplasm?

2) The authors show that the stress granules induced by mutant FUS expression are dissolved by treatment with their inhibitors, but the FUS inclusions seem resistant to the compounds. This could mean, as the authors propose, that the FUS inclusions are more stable than the stress granules. It could also mean that the FUS inclusions, contrary to expectation, do not directly coalesce with stress granules. Can the authors test this by pre-treating with their compounds before expressing mutant FUS? Does this treatment prevent FUS from forming cytoplasmic puncta?

Reviewer #2 (Comments to the Authors (Required)):

Freibaum et al describe the identification and characterization of two small molecule inhibitors (G3la and G3lb) of G3BP-driven stress granule assembly. They show that G3la and G3lb (and not their enantiomers) bind to G3BP1 and both block SG assembly and facilitate dissolution of pre-formed SGs. These small molecules affect SG formation in different cell types (U2OS and HeLa) and in response to different stresses: sodium arsenite and heat shock. Freibaum et al provide evidence that G3la and G3lb function by binding to G3BP1 and specifically interfering with the recruitment of caprin to the NTF2L domain of G3BP1. The authors also show that these small molecules can perturb SGs in a disease-relevant context, suggesting a potential avenue for

therapeutic development. This work is exciting and well-written, and the small molecule inhibitors of SG formation are likely to be of significant value to the condensate biology community. I have very few concerns about this work - see below.

Results/discussion:

- The author's claim that the effect of these compounds is long-lived would be better supported by an experiment in which cells are pre-treated with the compound over the time period used in other experiments (30 or 60 minutes), incubated for 24 hrs in the absence of inhibitor, and then assayed. This would show that the effects are long-lasting. The current data show that cells can tolerate the inhibitors for 24 hrs and maintain their effects on SG formation, but these data are insufficient to conclude that the effects are long-lasting.
- Paragraph 2 of the discussion suggests that the compounds do not affect translation. While I think it is unlikely that they do, the statement will be improved by showing this data. This would also strengthen the claim that the compounds are not toxic (Fig 2A).
- Most experiments were performed in GFP-G3BP stable cell lines. While I understand the utility of this cell line for live cell imaging, the effect of the compounds should also be shown in a WT cell line with endogenous G3BP1 to assure the validity and applicability of the conclusions.

Minor comments:

- Define acronyms/explain jargon: example, Fig 1E details, Fig 2A
- Supplementary Videos 9-24 not present with submitted files
- Unclear which fig number and legend corresponds to each supplemental video provided. Panels of videos are labeled with an inhibitor and concentration, but the figure legend just lists pre-incubation with one inhibitor at one concentration. Please clarify experimental procedure and figure legends associated with each video.
- Fig S5 B - difficult to see images, please enlarge.
- Discussion first paragraph: reference to Fig S1 should be S2; discussion paragraph 2 refers to Fig 1, but should be Fig 2 here.
- Discussion paragraph 2 states that G3la and G3lb leave the dimerization capability of G3BP intact, however the compounds bind to the NTF2L domain, which according to Sanders et al. 2020 and Yang et al. 2020 is dimerization domain. Please clarify.

Reviewer #1

Biomolecular condensation is an emerging principle that underlies many cellular processes. Stress granules are biomolecular condensates that form in the response to cellular stresses and have been connected to the pathogenesis of several diseases, including cancer and neurodegeneration.

The authors have previously discovered that two proteins, G3BP1 and G3BP2, are necessary and sufficient for stress granule formation. In this new paper, they rationally design small molecule G3BP1/2 inhibitors and show that they have potent activity against stress granules formed under diverse conditions. They also show that their compounds can either prevent the formation of stress granules or dissolve pre-existing ones and can do so in neurons. The compounds (but not their inactive enantiomers) can also block stress granule formation induced by mutant neurodegenerative disease proteins VCP and FUS. Importantly, these compounds exhibit little to no cellular toxicity.

This is a very exciting paper and the compounds presented here will be of immediate and broad interest to the cell biology field. The paper is very well written and the data are compelling. I recommend this paper be published in the JCB. I have some comments and suggestions for the authors to consider.

1) In this U2OS cell model, inducing stress granules using arsenite has been shown to also cause mislocalization of another ALS protein, TDP-43, from the nucleus to the cytoplasm. Does treatment with G3BP1/2 inhibitors reduce TDP-43 mislocalization to the cytoplasm?

Author response: As suggested by the reviewer, we have now examined whether our G3I compounds reduce the localization of TDP-43 to cytoplasmic stress granules. First, we used immunofluorescence to assess TDP-43 localization in U2OS cells treated with vehicle, inactive enantiomer, or active compound. We found that TDP-43 accumulates in stress granules in cells treated with vehicle, G3Ia' (inactive enantiomer), or G3Ib' (inactive enantiomer) but not G3Ia or G3Ib (inhibitor compounds) (**new Figure S2G, H**). Thus, as the reviewer anticipated, the G3BP1 inhibitors also prevent mislocalization of TDP-43 protein. To confirm this result, we used a CRISPR-modified U2OS cell line in which endogenous TDP-43 and G3BP1 were labeled with GFP and mRuby3, respectively. Again, we found that treatment with G3Ib (inhibitor), but not G3Ib' (inactive enantiomer), blocked the accumulation of TDP-43 into stress granules (**new Figure S2I, Supplemental Videos 5 and 6**).

2) The authors show that the stress granules induced by mutant FUS expression are dissolved by treatment with their inhibitors, but the FUS inclusions seem resistant to the compounds. This could mean, as the authors propose, that the FUS inclusions are more stable than the stress granules. It could also mean that the FUS inclusions, contrary to expectation, do not directly coalesce with stress granules. Can the authors test this by pre-treating with their compounds before expressing mutant FUS? Does this treatment prevent FUS from forming cytoplasmic puncta?

Author response: We have performed this experiment as suggested and found that pre-treatment with G3Ia and G3Ib inhibits the incorporation of G3BP1 into stress granules but does not affect the accumulation of mutant FUS into cytoplasmic puncta (**new Figure 5E, F**), suggesting that the formation of mutant FUS inclusions is independent of stress granules.

Reviewer #2

Freibaum et al describe the identification and characterization of two small molecule inhibitors (G3Ia and G3Ib) of G3BP-driven stress granule assembly. They show that G3Ia and G3Ib (and not their enantiomers) bind to G3BP1 and both block SG assembly and facilitate dissolution of pre-formed SGs. These small molecules affect SG formation in different cell types (U2OS and HeLa) and in response to different stresses: sodium arsenite and heat shock. Freibaum et al provide evidence that G3Ia and G3Ib function by binding to G3BP1 and specifically interfering with the recruitment of caprin to the NTF2L domain of G3BP1. The authors also show that these small molecules can perturb SGs in a disease-relevant context, suggesting a potential avenue for therapeutic development. This work is exciting and well-written, and the small molecule inhibitors of SG formation are likely to be of significant value to the condensate biology community. I have very few concerns about this work - see below.

Results/discussion:

- The author's claim that the effect of these compounds is long-lived would be better supported by an experiment in which cells are pre-treated with the compound over the time period used in other experiments (30 or 60 minutes), incubated for 24 hrs in the absence of inhibitor, and then assayed. This would show that the effects are long-lasting. The current data show that cells can tolerate the inhibitors for 24 hrs and maintain their effects on SG formation, but these data are insufficient to conclude that the effects are long-lasting.

Author response: We apologize for the lack of clarity. Our intention was to state that the effect of these compounds is durable and persists for the duration of time that the compound is present. We did not intend to suggest that the compound would be effective in the absence of the inhibitor – nor would we expect this to be the case, given the compounds' mechanism of action as a physical blocker of a G3BP1 binding site.

We have edited the text to clarify this point: the statement *"We also assessed whether the ability of these compounds to block stress granule formation in cells was long lived"* has now been edited to read, *"We also assessed whether the inhibitory effect of these compounds in dosed cell culture media persisted over an extended time period."*

- Paragraph 2 of the discussion suggests that the compounds do not affect translation. While I think it is unlikely that they do, the statement will be improved by showing this data. This would also strengthen the claim that the compounds are not toxic (Fig 2A).

Author response: We agree. We tested this idea by treating cells with compound/puromycin and using an antibody against puromycin to measure newly synthesized proteins. We found that the compounds do not alter translation in either stressed or unstressed cells. This data is now included in revised **Figure S4**.

- Most experiments were performed in GFP-G3BP stable cell lines. While I understand the utility of this cell line for live cell imaging, the effect of the compounds should also be shown in a WT cell line with endogenous G3BP1 to assure the validity and applicability of the conclusions.

Author response: As suggested by the reviewer, we assessed the impact of compounds on stress granule formation in a WT cell line with endogenous G3BP1. As shown in revised **Figure S2H**, both G3Ia and G3Ib (active inhibitors), not G3Ia' and G3Ib' (inactive enantiomers), effectively blocked stress granule formation as indicated by diffuse endogenous G3BP signal.

Minor comments:

- Define acronyms/explain jargon: example, Fig 1E details, Fig 2A

Author response: We have now added definitions for acronyms and jargon in the legends for Figures 1E and 2A.

- Supplementary Videos 9-24 not present with submitted files

Author response: We apologize for the confusion. This issue arose because of the automated transmission of our manuscript from bioRxiv to JCB. Both the bioRxiv and JCB manuscripts were provided to reviewers, but only the JCB manuscript included the full panel of supplementary videos.

- Unclear which fig number and legend corresponds to each supplemental video provided. Panels of videos are labeled with an inhibitor and concentration, but the figure legend just lists pre-incubation with one inhibitor at one concentration. Please clarify experimental procedure and figure legends associated with each video.

Author response: As in our response above, this issue arose because of automated transmission between bioRxiv and JCB. Only the JCB manuscript included the full panel of videos and figure legends.

- Fig S5 B - difficult to see images, please enlarge.

Author response: The revised manuscript now includes enlarged individual images in Figure S5.

- Discussion first paragraph: reference to Fig S1 should be S2; discussion paragraph 2 refers to Fig 1, but should be Fig 2 here.

Author response: These errors have been corrected in the revised manuscript.

- Discussion paragraph 2 states that G3la and G3lb leave the dimerization capability of G3BP intact, however the compounds bind to the NTF2L domain, which according to Sanders et al. 2020 and Yang et al. 2020 is dimerization domain. Please clarify.

Author response: Indeed, NTF2L is the dimerization domain. However, G3la and G3lb bind to an interaction surface that is on the opposite face of the dimerization surface; thus, compound binding by G3la and G3lb leaves dimerization intact. We have provided clarification on this point in the text.

December 11, 2023

RE: JCB Manuscript #202308083R

Dr. Joseph P Taylor
St. Jude Children's Research Hospital
Cell and Molecular Biology
262 Danny Thomas Place
Memphis, TN 38105

Dear Dr. Taylor:

Thank you for submitting your revised manuscript entitled "Identification of small molecule inhibitors of G3BP-driven stress granule formation". We would be happy to publish your paper in JCB pending final revisions necessary to meet our formatting guidelines (see details below).

A. MANUSCRIPT ORGANIZATION AND FORMATTING:

- 1) Text limits: Character count for Tools is < 40,000, not including spaces. Count includes abstract, introduction, results, discussion, and acknowledgments. Count does not include title page, figure legends, materials and methods, references, tables, or supplemental legends.
- 2) Figures limits: Tools may have up to 10 main text figures.
- 3) Figure formatting: Scale bars must be present on all microscopy images, including inset magnifications. * Molecular weight or nucleic acid size markers must be included on all gel electrophoresis. In order to accommodate readers with red-green color blindness, we suggest that you change all red/green color schemes.
- 4) Statistical analysis: Error bars on graphic representations of numerical data must be clearly described in the figure legend. The number of independent data points (n) represented in a graph must be indicated in the legend. Statistical methods should be explained in full in the materials and methods. For figures presenting pooled data the statistical measure should be defined in the figure legends. Please also be sure to indicate the statistical tests used in each of your experiments (either in the figure legend itself or in a separate methods section) as well as the parameters of the test (for example, if you ran a t-test, please indicate if it was one- or two-sided, etc.). Also, if you used parametric tests, please indicate if the data distribution was tested for normality (and if so, how). If not, you must state something to the effect that "Data distribution was assumed to be normal but this was not formally tested."
- 5) Abstract and title: The abstract should be no longer than 160 words and should communicate the significance of the paper for a general audience. The title should be less than 100 characters including spaces. Make the title concise but accessible to a general readership.
- 6) Materials and methods: Should be comprehensive and not simply reference a previous publication for details on how an experiment was performed. * Please provide full descriptions in the text for readers who may not have access to referenced manuscripts.
- 7) Please be sure to provide the sequences for all of your primers/oligos and RNAi constructs in the materials and methods. You must also indicate in the methods the source, species, and catalog numbers (where appropriate), along with the RRID for all of your antibodies. Please also indicate the acquisition and quantification methods for immunoblotting/western blots.
- 8) Microscope image acquisition: The following information must be provided about the acquisition and processing of images:
 - a. Make and model of microscope
 - b. Type, magnification, and numerical aperture of the objective lenses
 - c. Temperature
 - d. Imaging medium
 - e. Fluorochromes
 - f. Camera make and model

g. Acquisition software

h. Any software used for image processing subsequent to data acquisition. Please include details and types of operations involved (e.g., type of deconvolution, 3D reconstitutions, surface or volume rendering, gamma adjustments, etc.).

9) References: There is no limit to the number of references cited in a manuscript. * References should be cited parenthetically in the text by author and year of publication. Abbreviate the names of journals according to PubMed.

10) Supplemental materials: There are strict limits on the allowable amount of supplemental data. Tools may have up to 5 supplemental figures. Please also note that tables, like figures, should be provided as individual, editable files. A summary of all supplemental material should appear at the end of the Materials and methods section.

13) ORCID IDs: ORCID IDs are unique identifiers allowing researchers to create a record of their various scholarly contributions in a single place. Please note that ORCID IDs are now *required* for all authors. At resubmission of your final files, please be sure to provide your ORCID ID and those of all co-authors.

Please note that JCB now requires authors to submit Source Data used to generate figures containing gels and Western blots with all revised manuscripts. This Source Data consists of fully uncropped and unprocessed images for each gel/blot displayed in the main and supplemental figures. Since your paper includes cropped gel and/or blot images, please be sure to provide one Source Data file for each figure that contains gels and/or blots along with your revised manuscript files. File names for Source Data figures should be alphanumeric without any spaces or special characters (i.e., SourceDataF#, where F# refers to the associated main figure number or SourceDataFS# for those associated with Supplementary figures). The lanes of the gels/blots should be labeled as they are in the associated figure, the place where cropping was applied should be marked (with a box), and molecular weight/size standards should be labeled wherever possible.

Journal of Cell Biology now requires a data availability statement for all research article submissions. These statements will be published in the article directly above the Acknowledgments. The statement should address all data underlying the research presented in the manuscript. Please visit the JCB instructions for authors for guidelines and examples of statements at (<https://rupress.org/jcb/pages/editorial-policies#data-availability-statement>).

B. FINAL FILES:

**It is JCB policy that if requested, original data images must be made available to the editors. Failure to provide original images upon request will result in unavoidable delays in publication. Please ensure that you have access to all original data images prior

to final submission.**

Thank you for this interesting contribution, we look forward to publishing your paper in Journal of Cell Biology.

Sincerely,

Karla Neugebauer, PhD
Monitoring Editor

Andrea L. Marat, PhD
Senior Scientific Editor

Journal of Cell Biology

Reviewer #1 (Comments to the Authors (Required)):

The authors have addressed my comments and suggestions and the new data on TDP-43 and FUS are very interesting and will be broadly useful. I recommend publication in the JCB.